# COGFLOW: BRIDGING PERCEPTION AND REASONING THROUGH KNOWLEDGE INTERNALIZATION FOR VISUAL MATHEMATICAL PROBLEM SOLVING

**Shuhang Chen**[1]   **Yunqiu Xu**[1]   **Junjie Xie**[1]   **Aojun Lu**[3]   **Tao Feng**[4]
**Zeying Huang**[2]   **Ning Zhang**[2]   **Yi Sun**[2]   **Yi Yang**[1]   **Hangjie Yuan**[1*]
[1]Zhejiang University  [2]Intelligent Learning  [3]Sichuan University  [4]Tsinghua University
{sh.chen, jj.xie, yangyics, hj.yuan}@zju.edu.cn
{imyunqiuxu, fengtao.hi, futuretrader}@gmail.com
aojunlu@stu.scu.edu.cn   jzxjeff@163.com   sunyi@jzx100.cn

## ABSTRACT

Despite significant progress, multimodal large language models continue to struggle with visual mathematical problem solving. Some recent works recognize that visual perception is a bottleneck in visual mathematical reasoning, but their solutions are limited to improving the extraction and interpretation of visual inputs. Notably, they all ignore the key issue of whether the extracted visual cues are faithfully integrated and properly utilized in subsequent reasoning. Motivated by this, we present COGFLOW, a novel cognitive-inspired three-stage framework that incorporates a knowledge internalization stage, explicitly simulating the hierarchical flow of human reasoning: perception⇒internalization⇒reasoning. In line with this hierarchical flow, we holistically enhance all its stages. We devise Synergistic Visual Rewards to boost perception capabilities in parametric and semantic spaces, jointly improving visual information extraction from symbols and diagrams. To guarantee faithful integration of extracted visual cues into subsequent reasoning, we introduce a Knowledge Internalization Reward model in the internalization stage, bridging perception and reasoning. Moreover, we design a Visual-Gated Policy Optimization algorithm to further enforce the reasoning is grounded with the visual knowledge, preventing models seeking shortcuts that appear coherent but are visually ungrounded reasoning chains. Moreover, we contribute a new dataset MATHCOG for model training, which contains samples with over 120K high-quality perception-reasoning aligned annotations. Comprehensive experiments and analysis on commonly used visual mathematical reasoning benchmarks validate the superiority of the proposed COGFLOW. Project page: https://shchen233.github.io/cogflow/.

## 1 INTRODUCTION

Multimodal large language models (MLLMs) are rapidly advancing and have been applied across various vision–language applications (Peng et al., 2024b; Xu et al., 2024; 2025b; Yue et al., 2024; Liang et al., 2025; Jia et al., 2024a; Quan et al., 2024; Zhang et al., 2025c; Ma et al., 2024a;b). However, existing MLLMs continue to struggle with challenging visual mathematical problems, resulting in low answer accuracy and inconsistent reasoning chains. Some early attempts (Wang et al., 2025a; Shen et al., 2025) adopt a one-step reasoning framework that directly interleaves visual perception with reasoning in an unstructured manner, often resulting in both perceptual and reasoning errors. Another line of work (Chen et al., 2025a; Guo et al., 2025c) follows a decoupled reasoning pipeline that explicitly separates the perception and reasoning parts, with the former focusing on visual recognition and the latter responsible for subsequent inference. Yet in practice, we observe that such a pipeline often suffers from the reasoning drift issue, *i.e.*, it tends to yield illogical or unwarranted reasoning steps that disregard perceptual evidence (see Figure 1). These observations

---

*Corresponding author.

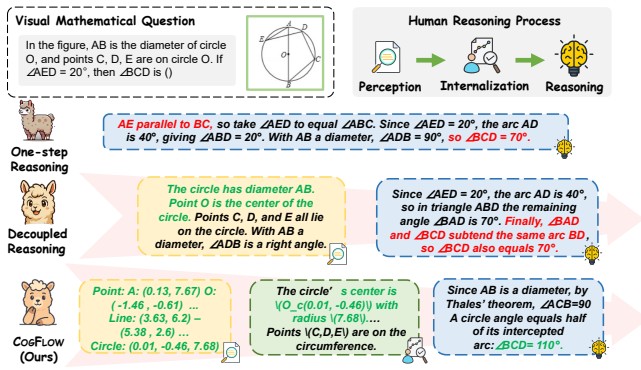 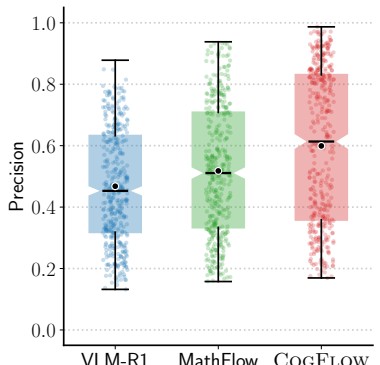

(a) *One-step* framework yields unstructured reasoning, while *decoupled* pipeline modularly disentangles the flow. We adopt a cognitive-inspired three-stage framework with knowledge internalization.

(b) Reasoning drift analysis across three representative pipelines, where higher precision indicates less reasoning drift.

Figure 1: The *one-step reasoning* framework (*e.g.*, VLM-R1 (Shen et al., 2025)) often yields suboptimal results, while the *decoupled reasoning* pipeline (*e.g.*, MathFlow (Chen et al., 2025a)) enhances perception yet still yields illogical reasoning steps that disregard visual evidence. In contrast, COGFLOW adopts a cognitive-inspired three-stage framework to effectively mitigate reasoning drift.

motivate the development of a new approach that not only achieves robust fine-grained recognition of mathematical visual elements (*e.g.*, diagrammatic primitives and symbols) but also faithfully incorporates extracted visual cues into subsequent reasoning.

Inspired by cognitive science findings on *knowledge internalization* (Ryan & Connell, 1989; Landy et al., 2014; Wu et al., 2022), this paper introduces **COGFLOW, a novel three-stage visual mathematical reasoning framework that better mirrors the typical hierarchical structure of the human reasoning process**. Concretely, after perception captures raw sensory input, an intermediate knowledge internalization stage transforms low-level perceptual signals into structured and semantically grounded knowledge representations (*e.g.*, humans internalize the perceptual facts that the line segment $AB$ is a diameter and the point $C$ lies on the circle into the knowledge that $\angle ACB = 90°$) before high-level reasoning begins. As illustrated in Figure 1a, to ensure both accurate extraction of visual information and its faithful use in reasoning, **COGFLOW explicitly models the hierarchical sequence of the human reasoning flow (*i.e.*, ❶perception⇒❷internalization⇒❸reasoning) and holistically enhances all three stages in synchrony with it**, where each improvement is tailored to the functional role of the corresponding stage in the human reasoning process.

Unlike prior approaches (Jia et al., 2024b; Chen et al., 2025a; Guo et al., 2025c) that decouple perception from reasoning trajectories (Ouyang et al., 2022) and enhance it with tailored tasks, **COGFLOW first integrates perception enhancement into a unified reinforcement learning (RL) framework through Synergistic Visual Rewards (SynVRs)**, enabling dynamic perception–reasoning interaction and improving generalization. Specifically, SynVRs complementarily optimize the model from two distinct perspectives: (1) a Visual Parameterized Reward (VPR) encoding normalized primitives (*i.e.*, points, lines, and circles) and calculating the Euclidean distance in a parameter space for precise and interpretable measurement; (2) a Visual Semantic Reward (VSR) that extracts semantic embeddings (Xie et al., 2025) from re-rendered images (derived from textual perception outputs) and measures the cosine distance in a semantic space to capture holistic style and layout consistency. Together, SynVRs ensure both local geometric fidelity and global perceptual coherence, forming trustworthy visual cues that serve as the foundation for effective visual mathematical reasoning.

Notably, despite progress in perception enhancement, all prior efforts (Jia et al., 2024b; Guo et al., 2025c; Wei et al., 2025) remain confined to accurate extraction of mathematical information from diagrams, while ignoring a key question: *are the extracted visual cues properly and faithfully integrated into subsequent reasoning?* As illustrated in Figure 1b, our empirical findings reveal a typical reasoning drift issue (*i.e.*, the reasoning stage in existing methods often deviates from perceptual results), leading to reasoning chains that appear coherent yet conflict with the underlying visual evidence. To prevent such drift and improve interpretability, **COGFLOW utilizes a Knowledge Internalization Reward (IntlzR) that bridges the perception and reasoning stages by encouraging**

**the model to generate structured and reasoning-ready outputs (*i.e.*, knowledge-internalized representations** (Ryan & Connell, 1989)) as a more reliable foundation for subsequent reasoning. Specifically, we curate positive trajectories integrating perception and reasoning processes with explicit internalization of perception primitives, and further derive five typical negative trajectories. Training with these trajectories enables the reward model to evaluate each response according to its fidelity to the internalized representation. IntlzR effectively improves the knowledge internalization stage, thereby reducing hallucinations and improving interpretability and robustness.

In accordance with the hierarchical flow of human reasoning, we further improve multi-step visual reasoning beyond enhanced perception and knowledge internalization. Existing approaches either follow a text-centric RL paradigm (Guo et al., 2025a) that is free from perceptual objectives (Chen et al., 2025b; Wang et al., 2025b), or overlook the structured dependency between perception and reasoning (Shen et al., 2025; Wang et al., 2025a). To ensure more stable reasoning in the presence of perceptual errors, **COGFLOW introduces Visual-Gated Policy Optimization (VGPO) strategy that explicitly anchors the reasoning process in perceptual accuracy**. In VGPO, a visual gate is designed to adaptively filter perceptual trajectories through perceptual quality assessment, retaining only high-quality ones before subsequent reasoning trajectory generation. If a low-quality perceptual trajectory is filtered out, the model regenerates alternative trajectories to obtain a higher-quality response. Along with the proposed visual gate, VGPO integrates an outcome-supervised Inference Reward (Shao et al., 2024) for optimization, further strengthening multi-step visual reasoning.

To facilitate research, we curate a new MATHCOG dataset for model training, which contains three subsets and over 120K samples with high-quality perception-reasoning aligned annotations. We conduct extensive experiments on commonly used visual math problem-solving benchmarks (Chen et al., 2025a; Zhang et al., 2024; Lu et al., 2024; Qiao et al., 2025a; Xiao et al., 2024; Zou et al., 2025) to comprehensively evaluate COGFLOW. The results show that COGFLOW consistently outperforms state-of-the-art MLLMs with comparable model sizes. Notably, it achieves on-par or even better results compared to advanced closed-source MLLMs with much larger model sizes. The main contributions of this paper can be summarized as follows:

- All prior works neglect whether extracted visual cues are faithfully used in reasoning. To address this issue, we present COGFLOW, a novel cognitive-inspired three-stage framework that faithfully simulates the hierarchical human reasoning flow: perception⇒internalization⇒reasoning.

- In line with human reasoning hierarchy, COGFLOW holistically enhances all three stages: SynVRs complementarily enhance accurate and complete diagram perception in parametric and semantic spaces; IntlzR improves the knowledge internalization ability for promoting faithful conversion of perceptual outputs into a canonical context used for subsequent inference; VGPO employs a visual gate to filter high-quality perception trajectories and enhances the stability of reasoning.

- To support model training, we curate a new dataset MATHCOG with disentangled perception and reasoning annotations. Comprehensive experiments on multiple visual mathematical benchmarks validate that COGFLOW achieves substantial gains in both answer accuracy and reasoning quality.

## 2 RELATED WORK

**Visual Mathematical Reasoning with MLLMs.** Solving visual mathematical problems (*e.g.*, geometry diagrams, algebraic plots, and *etc*) requires both strong reasoning ability and accurate interpretation of visual primitives and symbolic content (Yan et al., 2025; Qiao et al., 2025b; Zhang et al., 2025b). Most previous works are dedicated to improving the reasoning process, including chain-of-thought strategies (Xu et al., 2025a; Deng et al., 2024), tool-aided reasoning (Trinh et al., 2024; Chen et al., 2023b), test time scaling (Wang et al., 2025e; Hosseini et al., 2024), and reinforcement learning (Wang et al., 2025c; Jiang et al., 2025). Several recent works (Guo et al., 2025c; Jia et al., 2024b; Wei et al., 2024) suggest that one of the major bottlenecks in visual mathematical reasoning is inaccurate visual comprehension. They typically decouple perception from reasoning, and strengthen perception either by designing specialized visual encoders (Zhang et al., 2025a) or by introducing auxiliary visual tasks (Chen et al., 2025a). However, prior works ignore a key issue of whether correctly extracted visual cues are indeed faithfully incorporated into subsequent reasoning.

**Reinforcement Learning for Multimodal Reasoning.** Traditional actor–critic methods, such as proximal policy optimization (Yu et al., 2022), are computationally expensive. A lightweight alter-

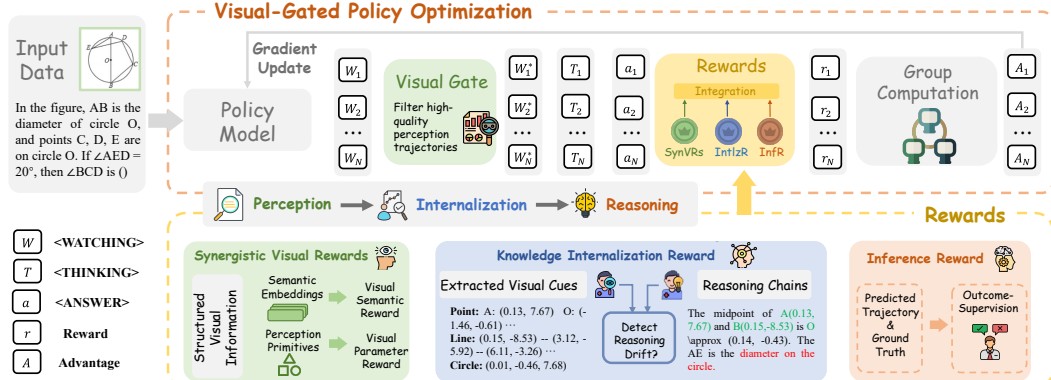

Figure 2: **Overview of the proposed visual mathematical reasoning framework COGFLOW.** Inspired by the canonical three-stage human reasoning flow, COGFLOW adopts a hierarchical pipeline that integrates Synergistic Visual Rewards (SynVRs) for enhanced perception, a Knowledge Internalization Reward (IntlzR) to bridge perception and reasoning, and Visual-Gated Policy Optimization (VGPO) with Inference Reward (InfR) to anchor multi-step reasoning in perceptual accuracy.

native is group relative policy optimization (GRPO) (Guo et al., 2025a), which stabilizes advantage estimation using group baselines. While its variants (Zheng et al., 2025; Yu et al., 2025) have been widely explored, GRPO has also been extended to multimodal reasoning (Wang et al., 2025c; Shen et al., 2025; Zhou et al., 2025). Some extensions introduce hybrid reward formulations augmented with preference signals during training (Wang et al., 2025c; Pan et al., 2025), whereas others propose two-stage multimodal reinforcement learning paradigms, *e.g.*, OVR (Wei et al., 2025). However, existing methods typically lack explicit mechanisms to strengthen the alignment between perception and reasoning, often leading to reasoning that is not firmly grounded in visual content.

## 3 COGFLOW: A COGNITIVE-INSPIRED HIERARCHICAL FRAMEWORK

Inspired by the typical cognitive process of human reasoning (*i.e.*, ❶perception⇒❷internalization ⇒❸reasoning), COGFLOW serves as a visual mathematical reinforcement learning framework that explicitly implements the internalization stage (see Figure 2). Before training, we first curate MATHCOG to support the subsequent training (see Figure 10). Specifically, the training pipeline of COGFLOW consists of two sequential phases: a supervised fine-tuning (SFT) phase and a reinforcement learning (RL) phase. The SFT phase endows the base model with initial visual perception and basic reasoning skills based on the MATHCOG-SFT dataset. During the RL phase, we optimize the policy based on the MATHCOG-RL dataset under the Visual-Gated Policy Optimization (VGPO) framework to explicitly anchor the reasoning process in perceptual accuracy. Concretely, VGPO introduces a visual gate to adaptively filter perceptual trajectories before reasoning trajectory generation. Furthermore, the rewards in VGPO are composed of three components: Synergistic Visual Rewards (SynVRs) for forming trustworthy perception, Knowledge Internalization Reward (IntlzR) for detecting reasoning drift and Inference Reward (InfR) for providing outcome-supervision.

### 3.1 FORMING TRUSTWORTHY PERCEPTION WITH SYNERGISTIC VISUAL REWARDS

COGFLOW first constructs Synergistic Visual Rewards (SynVRs), enabling dynamic perception-reasoning interaction and improving generalization. Specifically, the proposed SynVRs combine two complementary components: the Visual Parameterized Reward (VPR) and the Visual Semantic Reward (VSR), which respectively evaluate perceptual quality in the parametric and semantic spaces, thereby providing a synergistic perception feedback integrated into RL training loops.

**Measuring Perceptual Accuracy in Parameter Space.** As shown in Figure 3, VPR first converts structured visual information into parametric expressions. For example, the primitive Circle $(0.01, -0.46, 7.68)$ is transformed into the Equation $(x - 0.01)^2 + (y + 0.46)^2 = 7.68^2$.

We then compute the cost matrix $\mathcal{C}$ between GT Primitives $\mathcal{G}$ and predicted Primitives $\mathcal{P}$ and apply the Hungarian matching algorithm (Kuhn, 1955) to obtain the optimal one-to-one matching $\mathcal{H}$ that

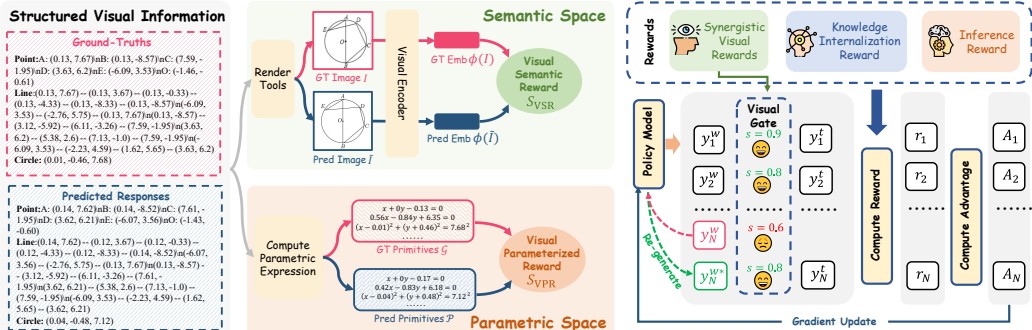

Figure 3: **Workflow of SynVRs.** SynVRs consist of a Visual Semantic Reward and a Visual Parametric Reward, ensuring local geometric fidelity and global perceptual coherence respectively. Together, these two complementary visual rewards provide a unified supervision mechanism for training robust and accurate visual perception.

Figure 4: **Pipeline of VGPO.** VGPO introduces visual gate and multiple rewards to strengthen multi-step visual reasoning. Coupling perceptual quality control with outcome-based optimization promotes stability.

minimizes the total cost $\mathcal{S}_{\mathrm{VPR}}$. The VPR offers interpretable, geometry-aware semantic supervision that avoids the pitfalls of pixel-level noise or black-box embedding similarity. More details are provided in Appendix §C.2.

**Capturing Holistic Layout and Style Consistency in Semantic Space.** We render the predicted response into an image $\tilde{I}$ and compare it against the ground-truth rendering $I$, using a frozen FG-CLIP encoder (Xie et al., 2025) $\phi(\cdot)$ with higher sensitivity to spatial features. The VSR score $\mathcal{S}_{\mathrm{VSR}}$ is the normalized cosine similarity. The higher values indicate closer agreement in global layout and style while preserving fine-grained geometric fidelity.

**Synergy of Complementary Visual Rewards.** Finally, the SynVRs score can be formulated as

$$\mathcal{S}_{\mathrm{SynVRs}} = \alpha \underbrace{\exp\left(-\frac{1}{|\mathcal{H}|}\sum_{(i,j)\in\mathcal{H}}\mathcal{C}(\mathcal{P}_i, \mathcal{G}_j)\right)}_{\mathcal{S}_{\mathrm{VPR}}} + (1-\alpha) \underbrace{\frac{1+\cos(\phi(\tilde{I}), \phi(I))}{2}}_{\mathcal{S}_{\mathrm{VSR}}}, \tag{1}$$

where $\alpha \in [0, 1]$ balances local geometric precision ($\mathcal{S}_{\mathrm{VPR}}$) and global visual consistency ($\mathcal{S}_{\mathrm{VSR}}$). $\mathcal{S}_{\mathrm{SynVRs}}$ has two roles: (1) it acts as a gate to prevent low-quality perceptions from propagating during policy generation, and (2) as a training reward during RL to encourage accurate perception.

## 3.2 Alleviating Reasoning Drift with Knowledge Internalization Reward

**Failure Modes in Knowledge Internalization.** Even with accurate perception, models often struggle to reliably internalize what they see. The empirical evidence suggests that such failures typically manifest in five forms: (1) *omit or misbind primitives*, losing essential elements or confusing their identifiers; (2) *introduce nonexistent facts*, fabricating geometric relations that are not present in the perceptual output; (3) *invoke external theorems inappropriately*, applying mathematical results that were never justified by the internalized structure; (4) *contradict geometric constraints*, producing inference steps that violate fundamental properties of the canonical representation; and (5) *refer inconsistently to established elements*, assigning shifting roles or properties to the same primitive across different reasoning steps. Therefore, to enhance knowledge internalization, it is crucial to mitigate these systematic failures.

**Rewarding Faithful Knowledge Internalization.** To address these systematic weaknesses, we introduce the Knowledge Internalization Reward (IntlzR), a trained reward model that evaluates whether each reasoning chain remains faithful to the internalized visual representation, thereby enforcing raw perception as the primary substrate of inference (*e.g.*, reasoning drift). Within COGFLOW, we train the IntlzR to enhance internalization explicitly, supported by the curated MATHCOG-IntlzR (see Appendix §C.3 for more details). This dataset is constructed from positive–negative pairs, with each positive matched to five corresponding negatives; positives are sampled from MATHCOG-SFT, and negatives are synthesized by injecting the five error types described

above, providing fine-grained supervision for distinguishing grounded from unanchored reasoning. To better leverage diverse negative signals in IntlzR and adaptively emphasize the most challenging trajectories while improving gradient efficiency and training stability, we adopt Softmax-DPO (Chen et al., 2024a) for optimization:

$$\mathcal{L}_{\text{Softmax-DPO}} = -\log \sigma \left( -\log \sum_{j=1}^{m} \exp\left(s_j^- - s^+\right) \right), \quad s = \beta\big[\log \pi_\theta(y \mid x) - \log \pi_{\text{ref}}(y \mid x)\big],$$
(2)

where $s^+$ denotes the score of the preferred trajectory, $\{s_j^-\}_{j=1}^m$ represent the corresponding scores of the dispreferred trajectories, $\beta$ is the KL penalty coefficient and $\sigma(\cdot)$ denotes the sigmoid function. This formulation contrasts one positive trajectory against multiple negatives simultaneously, thereby providing denser and more informative supervision, implicitly emphasizing hard negatives through softmax weighting, and yielding more stable optimization with stronger robustness to unseen misalignment patterns. During RL, IntlzR is evaluated stepwise and aggregated across the chain, rewarding trajectories in which the extracted visual cues are faithfully and adequately integrated into subsequent reasoning.

### 3.3 Strengthening Multi-Step Visual Reasoning with VGPO

Accurate perception and faithful internalization are necessary but not sufficient: the model must also produce long reasoning chains that are coherent, interpretable, and verifiably grounded. Hence, we introduce Visual-Gated Policy Optimization (VGPO), which integrates a visual gate with group-level optimization to regularize the reasoning process. Concretely, as shown in Figure 4, for an input question $x$ we sample $M$ candidate trajectories $y_i(x) = \big(y_i^{\text{w}}(x), y_i^{\text{t}}(x, y_i^{\text{w}}(x))\big)$. Here $y_i^{\text{w}}(x)$ denotes the *perception* trajectories, *i.e.*, structured parse of perceptual primitives and relations from the diagram, and $y_i^{\text{t}}(x, y_i^{\text{w}}(x))$ denotes the *reasoning* trajectories conditioned on $x$ and $y_i^{\text{w}}(x)$.

**Visual Gate for Reliable Visually Grounded Reasoning.** We introduce a visual gate that scores each perception against the visual evidence and forwards only the most faithful parse to reasoning. Specifically, given a perception candidate $y_i^{\text{w}}(x)$ and the ground-truth $\hat{y}^{\text{w}}(x)$, while $\tilde{I}_i$ and $I$ denote their respective renderings, we define the perceptual accuracy score as:

$$S_{\text{vis}}\big(y_i^{\text{w}}(x)\big) = \begin{cases} \mathcal{S}_{\text{VPR}}\big(y_i^{\text{w}}(x),\, \hat{y}^{\text{w}}(x)\big) + \mathcal{S}_{\text{VSR}}\big(\tilde{I}_i,\, I\big), & \text{training,} \\ \mathcal{S}_{\text{VSR}}\big(\tilde{I}_i,\, I\big), & \text{inference.} \end{cases}$$
(3)

Subsequently, the visual gate $\Gamma(\cdot)$ enforces perception quality by scoring each perception trajectory and accepting the first attempt whose score exceeds a preset threshold. If no attempt passes within $M$ trials, the gate returns the attempt with the highest $S_{\text{vis}}$:

$$\kappa = \begin{cases} \min\{\, k \in \{1, \ldots, M\} : S_{\text{vis}}(y_{i,k}^{\text{w}}(x)) \geq \tau \,\}, & \{k : S_{\text{vis}}(y_{i,k}^{\text{w}}(x)) \geq \tau\} \neq \varnothing, \\ M, & \{k : S_{\text{vis}}(y_{i,k}^{\text{w}}(x)) \geq \tau\} = \varnothing, \end{cases}$$
(4)

$$y_i^{\text{w}*}(x) = \Gamma\big(\{y_{i,k}^{\text{w}}(x)\}_{k=1}^{\kappa}\big) = \arg\max_{1 \leq k \leq \kappa} S_{\text{vis}}\big(y_{i,k}^{\text{w}}(x)\big),$$
(5)

where $\kappa$ is the stopping index, defined as the smallest $k$ whose score reaches acceptance threshold $\tau$, $y_i^{\text{w}*}(x)$ is the perception trajectory selected by the visual gate; and $y_{i,k}^{\text{w}}(x)$ is the $k$-th perception trajectory for input $x$.

**Stabilizing Multi-Step Reasoning via Visual-Gated Policy Optimization.** For each problem, we collect multiple candidate trajectories $y_i(x)$ and evaluate them under a group-level reward $r$ that integrates (1) the Synergistic Visual Rewards $R_{\text{SynVRs}}$ for perceptual fidelity, (2) the Knowledge Internalization Reward $R_{\text{IntlzR}}$ for internalization faithfulness, and (3) an Inference Reward $R_{\text{InfR}}$ capturing answer correctness and output format, given by:

$$r = \lambda_{\text{SynVRs}} R_{\text{SynVRs}} + \lambda_{\text{IntlzR}} R_{\text{IntlzR}} + \lambda_{\text{InfR}} R_{\text{InfR}}.$$
(6)

Then, the training optimization objective $\mathcal{L}$ is formulated as:

$$\mathcal{L} = -\mathbb{E}_i\Big[\min\big(\eta_i(\theta)A_i,\, \text{clip}(\eta_i(\theta),\, 1-\epsilon,\, 1+\epsilon)A_i\big)\Big] + \beta_{\text{KL}}\, \mathbb{D}_{\text{KL}}\big(\pi_\theta \,\|\, \pi_{\text{ref}}\big),$$
(7)

Table 1: Accuracy (%) and FlowVerse-style CoT-E (%) results on FlowVerse.

| Model | All | | Text Centric | | Text Limited | | Text Plus | | Vision Dense | | Vision Centric | | Vision Primary | |
|---|---|---|---|---|---|---|---|---|---|---|---|---|---|---|
| | CoT-E | Acc | CoT-E | Acc | CoT-E | Acc | CoT-E | Acc | CoT-E | Acc | CoT-E | Acc | CoT-E | Acc |
| Claude-3.5-Sonnet | 55.5 | 45.1 | 60.8 | 52.6 | 58.7 | 50.3 | 64.0 | 58.3 | 45.0 | 25.4 | 56.5 | 48.0 | 48.1 | 45.2 |
| GPT-4o | 56.9 | 49.7 | 61.0 | 56.8 | 58.7 | 54.4 | 62.2 | 58.2 | 45.2 | 30.0 | 58.6 | 52.6 | 54.1 | 51.0 |
| GPT-4V | 64.2 | 58.7 | 69.1 | 57.1 | 65.0 | 55.0 | 72.0 | 61.4 | 48.1 | 30.3 | 61.8 | 46.3 | 42.0 | 36.7 |
| MathFlow$^\star_{\text{GPT-4V}}$ | 64.2 | **59.5** | 69.5 | 58.2 | 67.2 | 57.4 | 71.1 | 64.1 | 52.7 | **47.5** | 62.1 | 57.1 | **60.4** | 57.0 |
| Gemini-2.5-pro | 64.5 | 56.2 | 68.3 | 61.9 | 66.1 | 60.8 | 68.9 | 64.1 | 52.1 | 37.1 | 65.7 | 57.9 | 57.0 | 54.6 |
| GPT-5 | **68.2** | 59.3 | **74.3** | **68.1** | **73.5** | **66.7** | **77.0** | **69.2** | **53.8** | 44.7 | **67.1** | **61.7** | 60.3 | **57.5** |
| InfiMM-Math-7B | 37.8 | 29.5 | 43.8 | 38.1 | 40.6 | 36.7 | 46.1 | 40.1 | 28.8 | 15.4 | 39.6 | 30.3 | 26.1 | 23.2 |
| InternVL2.5-8B | 46.3 | 40.1 | 49.2 | 41.3 | 40.5 | 38.4 | 49.6 | 42.7 | 38.4 | 20.2 | 41.0 | 35.9 | 35.8 | 33.9 |
| Math-LLaVA-13B | 39.3 | 30.8 | 45.1 | 39.3 | 44.4 | 37.2 | - | - | 36.2 | 18.6 | 41.7 | 35.9 | 37.0 | 34.2 |
| MultiMath-7B | 45.2 | 35.3 | 50.6 | 44.8 | 49.9 | 42.9 | - | - | 41.7 | 22.1 | 47.2 | 40.4 | 39.7 | 38.8 |
| SVE-Math-Qwen2.5-7B | 47.9 | 38.7 | 53.1 | 47.3 | 53.4 | 45.8 | - | - | 44.2 | 28.6 | 48.9 | 44.2 | 45.8 | 42.0 |
| VLM-R1-7B | 50.7 | 41.2 | 59.0 | 54.2 | 57.9 | 49.8 | 65.5 | 58.9 | 36.2 | 24.5 | 46.1 | 37.8 | 30.6 | 26.1 |
| CogFlow-7B | **66.0** | **56.2** | **67.9** | **58.6** | **67.3** | **58.3** | **68.1** | **60.9** | **57.8** | **42.7** | **68.2** | **61.1** | **66.7** | **63.5** |

Table 2: Accuracy (%) and MathVerse-style CoT-E (%) results on *testmini* set of MathVerse.

| Model | All | | Text Dominant | | Text Lite | | Text Only | | Vision Intensive | | Vision Dominant | | Vision Only | |
|---|---|---|---|---|---|---|---|---|---|---|---|---|---|---|
| | CoT-E | Acc | CoT-E | Acc | CoT-E | Acc | CoT-E | Acc | CoT-E | Acc | CoT-E | Acc | CoT-E | Acc |
| Qwen-VL-Plus | 21.3 | 11.8 | 26.0 | 15.7 | 21.2 | 11.1 | 25.2 | 14.5 | 18.5 | 9.0 | 19.1 | 13.0 | 21.8 | 10.0 |
| Gemini-Pro | 35.3 | 23.5 | 39.8 | 26.3 | 34.7 | 23.5 | 44.5 | 27.3 | 32.0 | 23.0 | 36.8 | 22.3 | 33.3 | 22.2 |
| Qwen-VL-Max | 37.2 | 25.3 | 42.8 | 30.7 | 37.7 | 26.1 | 47.9 | 28.9 | 33.6 | 24.1 | 35.9 | 24.1 | 35.9 | 21.4 |
| GPT-4V | 54.4 | 39.4 | 63.1 | **54.7** | 56.6 | 41.4 | 60.3 | **48.7** | 51.4 | 34.9 | 50.8 | 34.4 | 50.3 | 31.6 |
| MathFlow$^\star_{\text{GPT-4V}}$ | 56.7 | 43.8 | 65.2 | 51.1 | 58.9 | 46.4 | 62.1 | 48.5 | 53.7 | 40.3 | 52.1 | 37.4 | 52.5 | 39.0 |
| SPHINX-MoE-56B | 25.8 | 15.6 | 33.3 | 22.2 | 21.9 | 16.4 | 40.7 | 18.3 | 21.1 | 14.8 | 19.6 | 12.6 | 18.3 | 9.1 |
| InternLM-XC2-7B | 25.9 | 16.5 | 36.9 | 22.3 | 28.3 | 17.0 | 42.5 | 16.5 | 20.1 | 15.7 | 24.4 | 16.4 | 19.8 | 11.0 |
| Math-LLaVA-13B | - | 20.1 | - | 22.8 | - | 21.8 | - | - | - | 21.1 | - | 19.2 | - | 15.4 |
| MultiMath-7B | - | 26.9 | - | 34.8 | - | 30.8 | - | - | - | 28.1 | - | 25.9 | - | 15.0 |
| SVE-Math-Qwen2.5-7B | - | 31.4 | - | 37.6 | - | 36.8 | - | - | - | 34.9 | - | 31.5 | - | 16.0 |
| DVLR-14B | 48.1 | - | 54.3 | - | 49.0 | - | - | - | 46.3 | - | 47.2 | - | 43.8 | - |
| SophiaVL-R1-7B | 48.8 | - | 45.4 | - | 43.9 | - | - | - | 45.1 | - | 58.5 | - | **51.3** | - |
| CogFlow-7B | **53.9** | **39.5** | **60.7** | **41.9** | **51.2** | **37.0** | **52.3** | **40.1** | **55.0** | **42.4** | **58.7** | **44.8** | 44.2 | 26.3 |

$$y_i(x) = \big(y_i^{\text{w}}(x), y_i^{\text{t}}(x, \Gamma(y_i^{\text{w}}(x)))\big), \quad \eta_i(\theta) = \frac{\pi_\theta\big(y_i(x) \mid x\big)}{\pi_{\theta_{\text{old}}}\big(y_i(x) \mid x\big)}, \quad A_i = \frac{r_i - \mu_{\text{group}}(r)}{\sigma_{\text{group}}(r) + \varepsilon}, \quad (8)$$

where the $\mathbb{E}_i[\cdot]$ is the empirical average over candidates in the group; the $A_i$ is the group-normalized advantage; $\eta_i(\theta)$ is the likelihood ratio *w.r.t.* the behavior policy $\pi_{\theta_{\text{old}}}$; clip$(\cdot)$ uses the PPO-style hyperparameter $\epsilon$; $\mu_{\text{group}}$ and $\sigma_{\text{group}}$ are the group mean and standard deviation of $r_i$ (with a small stabilizer $\varepsilon$ in the denominator); $\beta_{KL}$ is the KL penalty coefficient and $\mathbb{D}_{\text{KL}}(\pi_\theta \| \pi_{\text{ref}})$ denotes the forward KL to a frozen reference policy (*e.g.*, the SFT model), implemented as the token/state-averaged KL in practice.

By combining visual gate with group-level optimization, VGPO stabilizes long-horizon training and encourages the emergence of interpretable chain-of-thoughts. This mechanism ensures that CogFlow converges toward the desired paradigm: *first perceive correctly, then reason correctly*.

## 4 EXPERIMENTS

### 4.1 EXPERIMENTAL SETUP

**Evaluation Benchmarks and Metrics.** We conduct experiments on widely used benchmarks (*i.e.*, FlowVerse (Chen et al., 2025a), MathVerse (Zhang et al., 2024), MathVista (Lu et al., 2024), We-Math (Qiao et al., 2025a), LogicVista (Xiao et al., 2024), and DynaMath (Zou et al., 2025)) with various visual mathematical reasoning tasks and different visual and textual complexity. Following common practice, we measure the accuracy (Acc) of the final answer and assess the reasoning ability from the intermediate reasoning process using the chain-of-thought evaluation (CoT-E).

**Baselines.** We compare our method with a wide range of closed-source MLLMs, such as GPT series (OpenAI, 2024; 2023; 2025), Gemini series (Team et al., 2025), DouBao-1.5-pro (Guo et al., 2025b), GLM-4.5V (Hong et al., 2025), Claude (Anthropic, 2024) and Qwen-VL-Plus (Bai et al., 2023), as well as open-source MLLMs, including InfiMM-Math (Han et al., 2025), In-ternVL2.5 (Chen et al., 2024b), InternLM-XC2 (Dong et al., 2024), Math-LLaVA (Shi et al.,

Table 3: **Accuracy (%) results on MathVista.** CogFLOW demonstrates consistent superiority.

| Model | All | FQA | GPS | MWP | TQA | VQA |
|---|---|---|---|---|---|---|
| GPT-4V | 49.9 | 43.1 | 50.5 | 57.5 | 65.2 | 38.0 |
| Claude-3.5-Sonnet | 67.7 | - | - | - | - | - |
| Doubao-pro-1.5 | **79.5** | **77.7** | **88.9** | **86.0** | **82.3** | **62.0** |
| G-LLaVA-7B | 25.1 | 19.1 | 48.7 | 3.6 | 25.0 | 28.7 |
| VCAR-7B | 33.7 | 30.9 | 34.6 | 38.7 | 37.3 | 28.5 |
| SPHINX-Plus-56B | 36.7 | 54.6 | 16.4 | 23.1 | 41.8 | 43.0 |
| SVE-Math-7B | 37.4 | 31.9 | 53.9 | 29.0 | 41.4 | 30.8 |
| MultiMath-7B | 50.0 | 40.1 | 66.8 | 61.8 | 50.0 | 33.0 |
| SophiaVL-R1-7B | 71.3 | - | - | - | 73.4 | - |
| ThinkLite-VL-7B | 71.6 | - | - | - | - | - |
| VL-Rethinker-7B | 73.7 | - | - | - | - | - |
| CogFLOW-7B | 76.8 | 70.4 | 93.1 | 73.7 | 86.9 | 59.3 |

Table 4: CogFLOW shows competitive accuracy (%) results on more visual math benchmarks.

| Models | WeMath | LogicVista | DynaMath |
|---|---|---|---|
| Claude-3.7-Sonnet | 49.3 | 58.2 | 39.7 |
| GLM-4.5V | 68.8 | 62.4 | 53.9 |
| Doubao-1.5-Pro | 65.7 | 64.2 | 44.9 |
| GPT-5 | 71.1 | 70.0 | **60.9** |
| Gemini-2.5-Pro | **78.0** | **73.8** | 56.3 |
| Ovis-8B | 27.2 | 39.4 | 20.4 |
| Qwen2.5-VL-8B | 35.2 | 44.1 | 21.0 |
| InternVL3-8B | 37.1 | 44.1 | 25.5 |
| Keye-VL-8B | 60.7 | 54.8 | 37.3 |
| InternVL3.5-8B | 57.0 | 57.3 | 37.7 |
| GLM-4.1V-9B | 63.8 | **60.4** | 42.5 |
| CogFLOW-7B | **64.1** | 58.1 | **46.2** |

2024), MultiMath-7B (Peng et al., 2024a), SVE-Math (Zhang et al., 2025b), VLM-R1 (Shen et al., 2025), MathFlow (Chen et al., 2025a), DVLR (Guo et al., 2025c), VCAR (Jia et al., 2024b), VL-Rethinker (Wang et al., 2025a), ThinkLite-VL (Wang et al., 2025d) and SPHINX-MoE (Liu et al., 2024a). Since some concurrent works have yet to release their models, we report the results provided in their original papers, some of which do not include category-wise performance.

**Implementation Details.** We initialize CogFLOW with Qwen2.5-VL-7B (Bai et al., 2025) and train it on our curated MATHCog dataset. For cold start, we first train CogFLOW on MATHCog-SFT subset for 2 epochs with a learning rate of $1 \times 10^{-5}$ and a batch size of 64. Subsequently, CogFLOW is optimized using VGPO on MATHCog-RL subset for 1 epoch with a learning rate of $1 \times 10^{-6}$ and a batch size of 16. The IntlzR reward model is based on Qwen2.5-VL-3B and trained on MATHCog-IntlzR subset for 3 epochs with a learning rate of $7 \times 10^{-6}$ and a batch size 64. All our models are trained on 16 NVIDIA A100 GPUs. Please refer to Appendix §C for more details.

## 4.2 MAIN RESULTS

As shown in Tables 1–4, CogFLOW achieves 66.0% accuracy on FlowVerse, 53.9% on MathVerse, 76.8% on Mathvista, 64.1% accuracy on WeMath and 46.2% on DynaMath, surpassing all open-source baselines by large margins. In terms of LogicVista, it delivers competitive performance, trailing only GLM-4.IV-9B. **CogFLOW's improvements in visual perception are particularly significant,** as evidenced by its superior performance on subsets where visual information dominates. For example, on FlowVerse it reaches 42.7% on Vision Dense, 55.6% on Vision Primary, and 61.1% on Vision Centric. On MathVerse, the gains are consistent across Vision Intensive (44.8%), Vision Dominant (42.1%), and Vision Only (25.7%). These results demonstrate that CogFLOW yields more accurate parsing of geometric primitives and relations, leading to stronger perception–reasoning integration. **CogFLOW also delivers higher-end-to-end problem-solving accuracy across all settings.** The largest absolute gains appear in visually demanding subsets, where perception-anchored reasoning is essential. This demonstrates that CogFLOW not only perceives diagrams more accurately but also reasons over them more effectively. Figure 19 further shows that CogFLOW produces more structurally accurate and semantically consistent predictions compared to strong baselines. Please refer to Appendix §D.12 for more detailed configurations.

## 4.3 ABLATION STUDIES

**Component Ablation.** Table 5 reports the ablation results of CogFLOW. We observe that every component in our framework contributes positively to overall performance, which confirms the necessity of jointly addressing perception, internalization, and reasoning. Among all modules, VGPO emerges as the most influential, as it directly stabilizes long-horizon reasoning through visual gate and group-level optimization, thereby yielding the largest single-module gain.

**Analysis on Synergistic Visual Rewards.** As shown in Figure 5, the model without any visual reward demonstrates reasonable performance. However, adding either VSR or VPR consistently

Table 5: **Ablation of three proposed components (*i.e.*, SynVRs, IntlzR and VGPO).** The visual gate is always enabled during inference.

| SynVRs | IntlzR | VGPO | FlowVerse | | MathVerse | |
|---|---|---|---|---|---|---|
| | | | CoT-E | Acc | CoT-E | Acc |
| ✗ | ✗ | ✗ | 57.4 | 48.7 | 48.2 | 35.6 |
| ✓ | ✗ | ✗ | 63.2 | 54.7 | 50.5 | 36.9 |
| ✗ | ✓ | ✗ | 62.7 | 53.5 | 49.9 | 36.2 |
| ✗ | ✗ | ✓ | 63.4 | 54.8 | 50.8 | 37.3 |
| ✓ | ✓ | ✗ | 64.4 | 55.1 | 52.1 | 38.0 |
| ✓ | ✓ | ✓ | **66.0** | **56.2** | **53.9** | **39.5** |

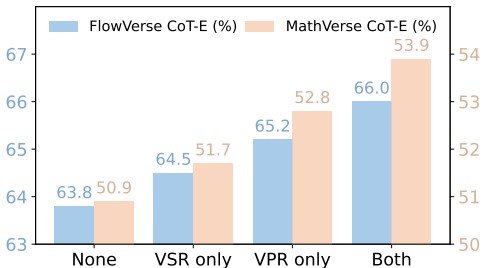

Figure 5: **Ablation analysis of SynVRs.** Variants exhibit consistent improvements.

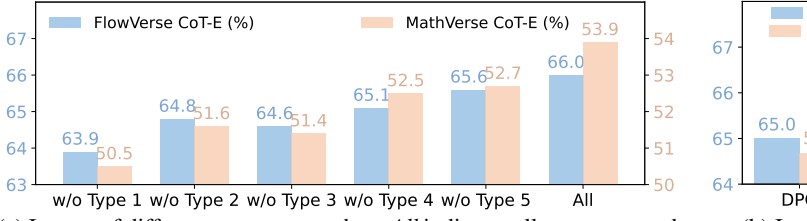

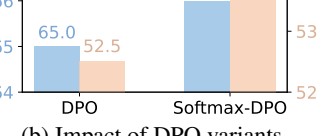

(a) Impact of different error types, where *All* indicates all types are used.

(b) Impact of DPO variants.

Figure 6: Ablation analysis of the proposed Knowledge Internalization Reward.

improves both CoT-E and final accuracy across MathVerse and FlowVerse. The best results come from combining VSR and VPR, improving CoT-E by up to +3.0% and accuracy by +1.7% on MathVerse, and CoT-E by +2.2% and accuracy by +2.1% on FlowVerse over the baseline. These results confirm that VPR ensures geometric precision while VSR stabilizes the global layout, and their combination provides the most reliable perceptual supervision.

**Analysis on Knowledge Internalization Reward.** As shown in Figure 6, removing any single Error type consistently degrades performance, indicating that each contributes complementary supervision for internalization. The largest drops arise when excluding *omission/misbinding primitives* or *contradicting geometric constraints*, highlighting that correctly binding primitives and respecting core geometric constraints are most critical for keeping reasoning tied to perception. Figure 6 compares different training strategies during training. Vanilla DPO (Rafailov et al., 2023) with all categories offers moderate gains, whereas Softmax-DPO achieves the best results, reaching 66.0% / 56.2% on FlowVerse. This suggests that fine-grained, within-problem preference modeling better captures the nuances of perception-grounded reasoning.

**Analysis on Visual-Gated Policy Optimization.** As shown in Figure 7, the distribution of visual reward values shifts steadily upward among different post-training methods, with VGPO producing both higher medians and more concentrated high-reward samples. This indicates that VGPO further enhances stability and fidelity by explicitly introducing a visual-gated mechanism. To disentangle the effect of VGPO from the effect of the gate itself, we conducted an additional ablation of the visual gate, as shown in Figure 8. These results indicate that visual gate is also beneficial during inference, yielding a consistent absolute gain of around 0.6-1% accuracy even when the model is trained without VGPO. Moreover, VGPO provides an additional and more substantial improvement, because it uses the visual gate during training to shape the policy itself, rather than only filtering outputs at test time.

**Error Type Analysis.** As shown in Figure 9, we employ GPT-5 (OpenAI, 2025) to classify each response on the FlowVerse benchmark into one of four categories: Perception Error, Knowledge Internalization Error, Reasoning Error, and Correct. The first row covers specialized visual-math MLLMs (*i.e.*, MultiMath-7B (Peng et al., 2024a) and SVE-Math-7B (Zhang et al., 2025b)), an GRPO-style model (*i.e.*, VLM-R1 (Shen et al., 2025)), the decoupled method (*i.e.*, MathFlow-7B (Chen et al., 2025a)), and GPT-4o (OpenAI, 2024); the second row traces our variants from the SFT+GRPO baseline through +SynVRs, +IntlzR, and +VGPO to COGFLOW. We observe: (1) GPT-4o's strong overall performance is driven by superior reasoning, yet its diagram perception lags behind specialized 7B models, indicating that strengthening perception and internalization remains essential; (2)

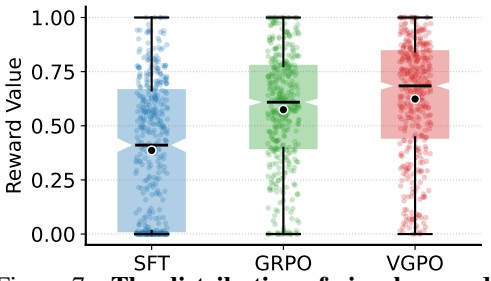

Figure 7: **The distribution of visual reward values among different post-training methods.** A higher concentration of values indicates stronger perceptual grounding achieved by the corresponding training strategy.

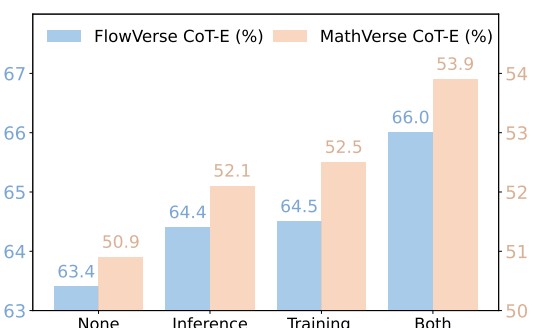

Figure 8: **Ablation analysis of visual gate.** *Training* and *Inference* indicates visual gate is only used in training and inference phases respectively.

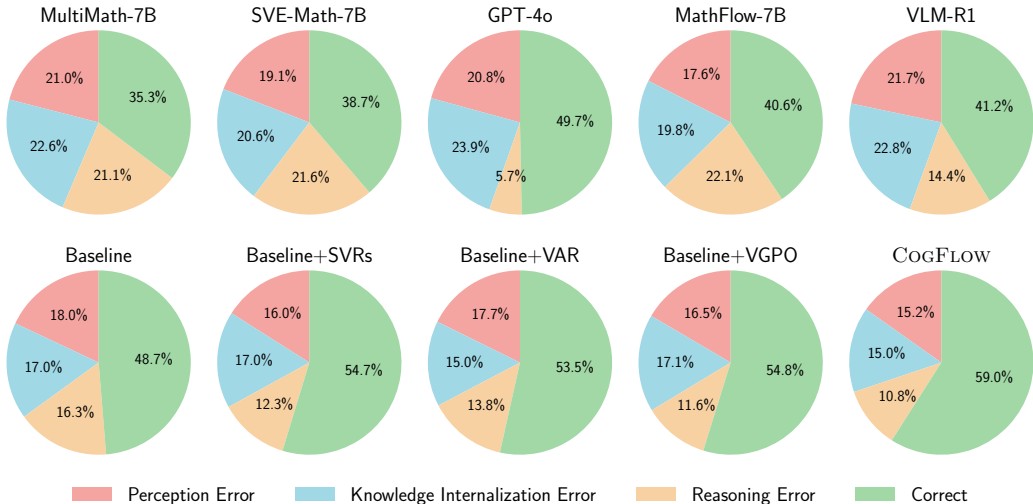

Figure 9: **Error-type analysis.** We analyze error-type distributions for COGFLOW variants alongside specialized visual–math models, the GRPO-style model, and the decoupled method. The baseline is denoted by the SFT+GRPO setting.

although methods such as MathFlow-7B and VLM-R1 reduce Perception Error and Reasoning Error to some extent, these methods do not meaningfully alleviate Knowledge Internalization Error; (3) +SynVRs chiefly reduces Perception Errors ($-2\%$), confirming that geometry-aware and semantic visual rewards improve perceptual fidelity; (4) +IntlzR primarily mitigates reasoning drift, reducing Knowledge-Internalization Error by $-2\%$ relative to the baseline; and (5) when all components are combined, COGFLOW minimizes every error type while maximizing the proportion of correct predictions. Furthermore, we provide a systematic case study in Appendix §D.13 to analyze the error compensation mechanism.

## 5 CONCLUSION

Faithful integration of visual perception into reasoning remains a critical yet overlooked challenge in visual mathematical problem solving. In this paper, we observe a prevalent reasoning drift issue and propose a cognitive-inspired three-stage framework COGFLOW that explicitly models the hierarchical human reasoning flow from perception to knowledge internalization and finally to reasoning. By jointly enhancing perceptual fidelity, enforcing structured knowledge internalization, and anchoring multi-step reasoning to visual evidence, COGFLOW effectively mitigates reasoning drift and improves both reasoning accuracy and interpretability. Extensive experiments on commonly used visual mathematical benchmarks validate its consistent advantages over existing MLLMs.

ACKNOWLEDGMENTS

This work was supported in part by the National Natural Science Foundation of China (62402432, 62533019) and in part by the Fundamental Research Funds for the Central Universities (226-202500080).

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

# APPENDIX

## TABLE OF CONTENTS

# A  GENERAL DISCUSSIONS

## A.1  ETHICS STATEMENT

Our study focuses on visual mathematical reasoning and does not involve human-subjects experiments, personally identifiable information, or biometric data. All benchmarks used are publicly available under their original licenses. We introduce MATHCOG (SFT/IntlzR/RL subsets) to support training. MATHCOG contains diagrammatic math items and corresponding solutions (including synthetic negatives generated by controlled perturbations of reasoning), and—to the best of our knowledge—contains no sensitive or personal data. When reusing third-party content, we preserve original attribution, respect redistribution constraints, and release only the metadata/derivatives permitted by the source licenses. We do not identify any foreseeable ethical risks associated with this work. We disclose computing resources and carbon-related considerations in the appendix and follow common practices to reduce the footprint (mixed precision, batching, and early stopping). The authors declare no conflicts of interest beyond those listed on the title page.

## A.2  REPRODUCIBILITY STATEMENT

To facilitate reproducibility, we have released the full training and evaluation package for COGFLOW, including code, configuration files, and training dataset.

- **Data.** We have released our constructed MATHCOG dataset (including MATHCOG-SFT, MATH-COG-IntlzR, and MATHCOG-RL subsets). We have also provided the construction scripts for MATHCOG-IntlzR, including the positive–negative pairing procedure and the five error-type perturbations used to synthesize negative trajectories.

- **Training and Implementation.** We have provided runnable training recipes for the two-stage pipeline (SFT followed by VGPO-based RL), including the base model initialization, optimizer settings, batch sizes, learning rates, training epochs, and hardware requirements. We have released the exact reward formulation and hyperparameters used by VGPO, including the reward weights $(\lambda_{\mathrm{SynVRs}}, \lambda_{\mathrm{IntlzR}}, \lambda_{\mathrm{InfR}})$, the SynVRs mixing coefficient $\alpha$, and the visual gate parameters.

- **Evaluation Protocol.** We have released evaluation scripts for all benchmarks used in the paper and report metrics consistent with the main text (answer accuracy and CoT-based evaluation). For baselines whose models or code are not publicly available, we have clearly indicated which numbers are taken from the original papers.

- **Ablations and Diagnostics.** We provide detailed ablation studies in §4.3, covering the contributions of SynVRs, IntlzR, VGPO, and the visual gate. We further report extended analyses of VGPO and the visual gate in §C.6. All key training hyperparameters and configurations are specified in §C, and the visual gate parameters are defined in §3.3.

## A.3  THE USE OF LARGE LANGUAGE MODELS

A large language model (*i.e.*, ChatGPT-5) was used in two ways during the preparation of this paper:

- To aid in polishing the writing, including improvements to grammar, clarity, and readability.
- For retrieval and discovery, such as identifying and organizing related prior work.

The model did not contribute to research ideation, experiment design, data analysis, or interpretation of results. Its use was limited to non-scientific assistance (*e.g.*, language polishing, rephrasing, and formatting) under direct author supervision. All scientific content, methodological decisions, and conclusions are the sole responsibility of the authors.

## A.4  LIMITATIONS AND FUTURE WORKS

While our approach demonstrates strong performance, it has certain limitations. Most notably, the training procedure is computationally demanding, requiring substantial resources to achieve stable improvements. This dependence on large-scale computation may hinder broader adoption, especially in resource-constrained settings.

For future work, we would like to expand our method (especially the method of obtaining visual primitives) to general scenes. While this paper is primarily focused on visual mathematical problems, the procedure for obtaining these primitives is domain-agnostic. The notion of primitives, as a structured representation of visual information, naturally extends to natural scenes beyond the domain of shapes. Concretely, for natural images we could first leverage detection or segmentation models to obtain instance-level regions and their spatial extents. With the assistance of an LLM and lightweight human verification, these outputs can be normalized into a unified primitive schema that encodes structured semantic features for each instance and their relationships. Such a representation enables a consistent knowledge-internalization step that transforms raw perceptual outputs into a compact, reasoning-ready state, allowing downstream modules to operate over explicit and verifiable visual evidence.

COGFLOW's strong performance on visual mathematical reasoning suggests that the pipeline generalizes beyond visual math problems. By explicitly decomposing the process into perception, knowledge internalization, and reasoning—and training these stages to remain mutually consistent—we obtain a broadly effective framework for visually grounded multimodal reasoning, with the potential to achieve strong performance when transferred to other vision–language domains.

# B    ADDITIONAL DETAILS OF MATHCOG DATASET

Existing multimodal reasoning corpora provide abundant natural language annotations but rarely disentangle perception from reasoning (Zhang et al., 2025a; Gao et al., 2025). This gap makes it difficult to supervise models in a way that enforces accurate perception and strengthens their ability to internalize visual content. To address this, we construct the MATHCOG dataset, which explicitly separates the *watching* (perception) and *thinking* (reasoning) stages. Furthermore, to support the different training phases of COGFLOW, we curate three tailored subsets: MATHCOG-SFT, MATH-COG-IntlzR, and MATHCOG-RL.

## B.1    DATASET LICENSE AND INTENDED USE

The MATHCOG dataset will be released under the `CC-BY-4.0` license, which permits redistribution and adaptation for both academic research and commercial use, provided that appropriate credit is given.

The dataset is intended primarily for research on multimodal reasoning, visual perception, and mathematical problem solving. It is designed to facilitate studies on perception–reasoning alignment in multimodal large language models. The dataset should *not* be used for purposes unrelated to research or education, including surveillance, profiling, or decision-making in sensitive domains such as healthcare or law enforcement. We encourage responsible use and proper citation in all derivative works.

## B.2    DATA CURATION

### B.2.1    DATA COLLECTION

To construct MATHCOG, we first collect a large pool of geometry-related visual math problems from existing corpora, including MAVIS (Zhang et al., 2025a), Geo170K (Gao et al., 2025), and LLaVA-CoT (Xu et al., 2025a). We then perform a careful filtering process to obtain a high-quality subset of 111,752 problems. Specifically, we remove (1) problems dominated by weak geometric relevance charts (*e.g.*, histograms, pie charts) or artistic illustrations that deviate from geometric reasoning, and (2) diagrams with either excessively low resolution ($< 64$) or overly high resolution ($> 2048$), which would otherwise compromise model training consistency.

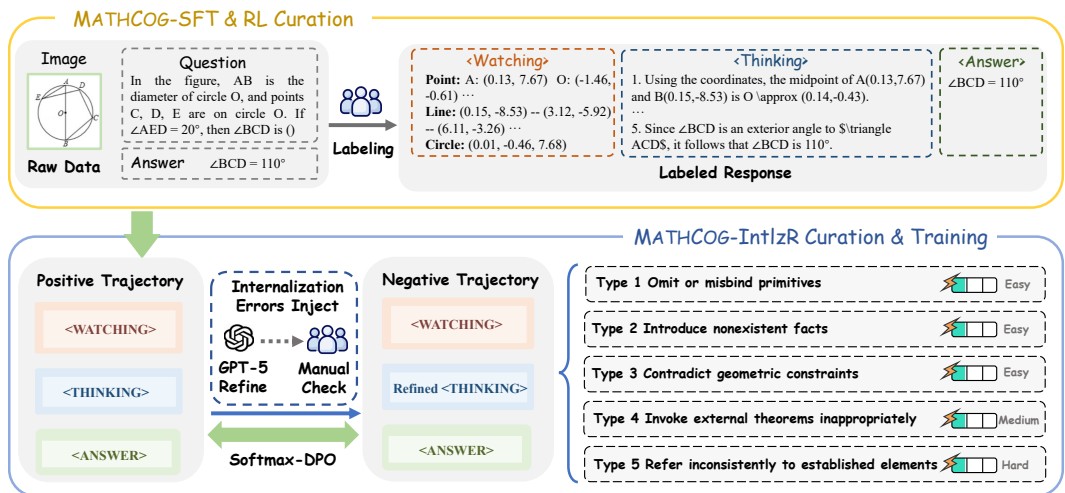

Figure 10: **Curation pipeline of MATHCOG.** We label MATHCOG-SFT and MATHCOG-RL based on the raw data. To construct MATHCOG-IntlzR, we sample positive examples from MATHCOG-SFT and generate five typical negative examples. Then, we adopt Softmax-DPO to train the IntlzR model.

Table 6: Prompt configuration for refining the MATHCOG dataset.

> **System:**
> *# Your Role: expert math geometry teacher*
>
> *## Objective*
> *You will be provided with a visual mathematics problem, along with its perception output (`<WATCHING>`) and a raw solution (`<THINKING>`).*
> *Your task is to refine the solution such that the reasoning process explicitly internalizes the perceptual information before carrying out logical inference.*
>
> *## Output Format*
> *The output format should strictly follow the example following:*
> *– Question: XXX*
> *– `<WATCHING>`: XXX*
> *– `<THINKING>`: XXX*
> *– Refined `<THINKING>`: XXX*
>
> *Now, here is the data you need to refine:*
> *– Question: {question}*
> *– `<WATCHING>`: {watching}*
> *– `<THINKING>` (raw): {solution}*
> *– Refined `<THINKING>`:*

### B.2.2 MATHCOG-SFT AND MATHCOG-RL CURATION

**Visual Primitive Annotation.** To construct a perception-enhanced dataset, we developed a preprocessing pipeline as illustrated in Figure 10. Since the raw dataset lacks sufficient perceptual information, we first curate additional geometric details from the original data. Formally, let $I$ denote the raw image. We apply OpenCV-based operators to extract a set of primitive geometric elements:

$$E = \text{Extract}(I; \theta_{\text{cv}}), \tag{9}$$

where $\theta_{\text{cv}}$ represents the parameters of the extraction process.

We utilize the OpenCV library to detect pixel-coordinate-based visual primitives in the images from the collected data, including endpoints and intersections of line segments, as well as circular features. For circles, we directly extract center coordinates $(x, y)$ and radius $r$ in pixel space. For points (*e.g.*, vertices and character annotations), we similarly obtain precise pixel coordinates and establish optimal correspondences between detected visual elements and character labels via the Hungarian algorithm, thereby obtaining final mappings such as $A : (x_1, y_1)$ and $B : (x_2, y_2)$.

To unify the representation, the extracted elements are normalized and remapped to a fixed coordinate system, following the normalization scheme in prior work (Wei et al., 2024):

$$\hat{E} = \text{Normalize}(E) \times 20 - 10, \tag{10}$$

where normalization is performed by dividing coordinates by the image width and height, and the subsequent linear transformation maps the results into the range $[-10, 10]$.

The complete pipeline can thus be summarized as:

$$\tilde{E} = \text{Annotate}(\text{Normalize}(\text{Extract}(I; \theta_{\text{cv}})) \times 20 - 10; \mathcal{H}). \tag{11}$$

This ensures that the final dataset not only captures essential geometric primitives but also provides high-quality, standardized representations suitable for subsequent training and evaluation, thus forming the perception part.

**Reasoning Annotation.** Based on these visual primitives, we further construct reasoning trajectories from the solution rationales in the collected data. The goal is to (i) explicitly internalize the structured primitives into a symbolic representation, and (ii) perform step-by-step logical inference over this internalized state to answer the question. These trajectories are first generated with the assistance of GPT-5 and are subsequently verified and refined by human annotators. The prompt is shown in Table 6. Finally, MATHCOG-RL is created by sampling 10,000 examples from this dataset, while MATHCOG-SFT is generated by sampling 100,000 examples.

A dedicated team of 31 professional annotators carried out this quality-assurance protocol over a one-month period, ensuring high-fidelity and consistent ground-truth annotations.

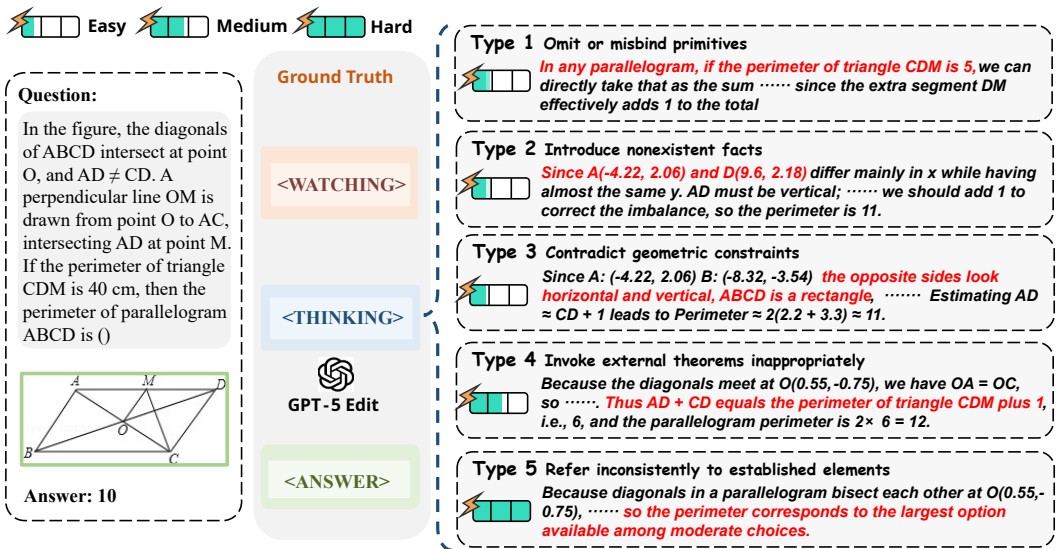

Figure 11: The illustration of data for Knowledge Internalization Reward.

### B.2.3 MATHCOG-INTLZR CURATION

Building on MATHCOG-SFT, we construct MATHCOG-IntlzR as a set of contrastive reasoning trajectories with explicit perturbations:

We first sample 10,000 examples as positive samples from MATHCOG-SFT for MATHCOG-IntlzR. As mentioned earlier, we categorize the errors into five types: (1) omitting or misbinding primitives, (2) introducing nonexistent facts, (3) contradicting geometric constraints, (4) invoking external theorems inappropriately, and (5) referring inconsistently to established elements. The negative data for MATHCOG-IntlzR is constructed based on these five error types. The curation of MATHCOG-IntlzR is illustrated at the bottom of the Figure 10. We first use an LLM, such as GPT-5 (OpenAI, 2025), to generate negative trajectories conditioned on each positive trajectory by explicitly instructing it to modify the corresponding "Thinking" part, as specified in the prompt shown in Table 7. The generated candidates are then manually checked. For example, for the *Type 1 error: Omit or misbind primitives* in Figure 11, we input the designed prompt into the LLM and obtain the variant "In any parallelogram, if the perimeter of triangle CDM is 5, . . . ", which deliberately alters the relevant content in the "Thinking" part of the original positive example. Specifically, for every positive trajectory we generate five distinct negative counterparts, resulting in 10,000 positive and 50,000 negative trajectories for Softmax-DPO training.

The easy/medium/hard labels characterize the relative difficulty for the model to detect and correct different types of reasoning drift errors, rather than the difficulty of constructing these cases. Easy errors are typically local and isolated (*e.g.*, a minor mistake in a single visual primitive or reasoning step) and can often be corrected using limited contextual information. Medium errors involve multiple interacting primitives or steps and require the model to integrate information across a broader context. Hard errors are globally entangled with the full reasoning chain, where correcting them demands a coherent understanding of both the visual configuration and the multi-step logical structure of the solution.

### B.3 DATASET STATISTICS

Table 8 provides the statistics for the MATHCOG dataset, detailing the number of problems and the distribution of data across different categories. The dataset consists of a total of 121,730 problems, with 100,000 labeled for supervised fine-tuning (MATHCOG-SFT) and 10,000 labeled for reinforcement learning (MATHCOG-RL). Additionally, there are 10,000 positive samples and 50,000 negative samples used in the MATHCOG-IntlzR (Visual-Augmented Reward) subset. A validation set contains 1,730 examples.

Table 7: Prompt for error type generation in MATHCOG-IntlzR.

**Omit or Misbind Primitives:**
*Starting from the correct reasoning, deliberately omit or misbind at least one basic geometric primitive (points, lines, circles, etc.), while keeping the overall reasoning fluent and locally plausible.*
*– Positive Data: {positive data}*

**Introduce Nonexistent Facts:**
*Starting from the correct reasoning, introduce at least one geometric or numerical fact that is not supported by the given figure or problem statement, but make the reasoning appear locally coherent and natural.*
*– Positive Data: {positive data}*

**Contradict Geometric Constraints:**
*Starting from the correct reasoning, modify the chain of thought so that at least one step violates the true geometric constraints (e.g., equal lengths, parallelism, angle measures), while preserving a seemingly reasonable narrative.*
*– Positive Data: {positive data}*

**Invoke External Theorems Inappropriately:**
*Starting from the correct reasoning, inappropriately invoke at least one external theorem or formula whose preconditions are not satisfied, or whose use is not justified by the internalized structure, while keeping the explanation linguistically smooth.*
*– Positive Data: {positive data}*

**Refer Inconsistently to Established Elements:**
*Starting from the correct reasoning, alter the chain of thought so that references to previously established elements (points, lines, relationships) become inconsistent, such as swapping labels or changing properties across steps, but without breaking the overall fluency of the text.*
*– Positive Data: {positive data}*

Table 8: Statistics of MATHCOG.

| Statistic | Number |
|---|---|
| Total Problem | 121,730 |
| - Number of MATHCOG-SFT | 100,000 |
| - Number of MATHCOG-RL | 10,000 |
| - Number of positive data of MATHCOG-IntlzR | 10,000 |
| - Number of negative data of MATHCOG-IntlzR | 50,000 |
| - Number of data for validation | 1730 |
| Maximum question length | 780 |
| Maximum watching length | 1,322 |
| Maximum thinking length | 1,288 |
| Average question length | 186 |
| Average watching length | 463.03 |
| Average thinking length | 527.79 |

Figure 12 shows the length of the various components. The maximum question length is 780, while the maximum lengths for the watching and thinking components are 1,322 and 1,288, respectively. On average, the question length is 186, with the watching and thinking components averaging 463.03 and 527.79, respectively.

### B.4 DATASET EXAMPLES

Figure 13 illustrates three representative samples from the MATHCOG dataset (covering both SFT and RL subsets), explicitly demonstrating our unique data structure that disentangles perception from reasoning. Each sample is organized into four distinct components: the raw visual input (`<IMAGE>`), the problem statement (`<PROBLEM>`), the perception trajectory (`<WATCHING>`), the reasoning trajectory (`<THINKING>`), and the final answer (`<ANSWER>`).

Specifically, the perception trajectory translates raw diagrammatic pixels into structured geometric primitives (Points, Lines, and Circles) with precise coordinates in a visual parameterized space. For instance, in the leftmost example involving a rectangle, the model identifies specific coordinates for vertices $A$ and $B$ (*e.g.*, $A : (-7.56, 3.5)$) and the intersection point $O$. Similarly, in the center

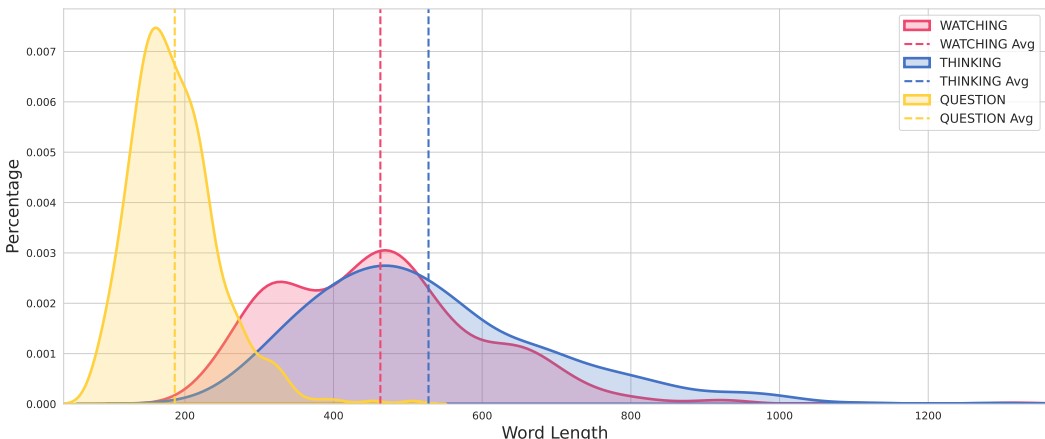

Figure 12: **Distribution of Word Length among QUESTION, WATCHING, THINKING.** We present the distribution of word length, with the horizontal axis representing word length and the vertical axis depicting the corresponding probability distribution.

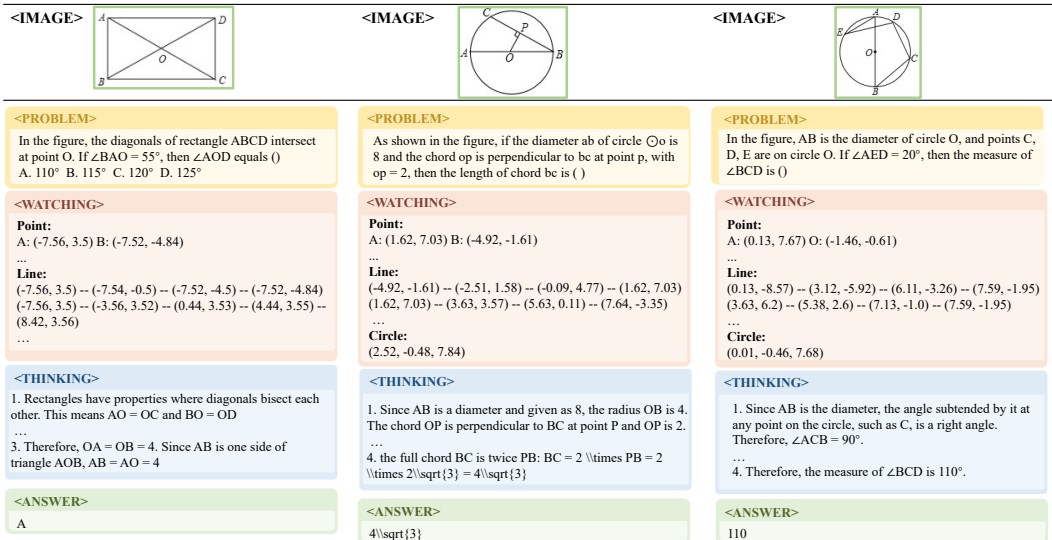

Figure 13: Representative cases in the MATHCOG-SFT and MATHCOG-RL.

and right examples, the model explicitly parameterizes circles by identifying center coordinates and radii, *e.g.*, Circle: $(2.52, -0.48, 7.84)$.

The reasoning trajectory section then performs logical inference anchored in these extracted visual cues. For example, in the rightmost case, the reasoning process explicitly links the perceived fact that "AB is the diameter" (derived from the coordinates in `<WATCHING>`) to the geometric theorem that the angle subtended by a diameter is $90°$, leading to the correct deduction that $\angle ACB = 90°5$. This structured format ensures that the final `<ANSWER>` is derived from a cognitive flow of perception$\Rightarrow$ internalization$\Rightarrow$reasoning.

# C  ADDITIONAL IMPLEMENTATION DETAILS

## C.1  MODEL INITIALIZATION

As shown in Table 9, we initialize COGFLOW with a 7B-scale pretrained MLLM backbone, Qwen2.5-VL-7B (Bai et al., 2025). All parameters of the vision encoder, perception modules, and language decoder are updated during training without freezing any component.

## C.2  THE DETAILS OF VPR

The Visual Parameterized Reward (VPR) metric quantifies perceptual accuracy within the parameter space of geometric primitives. As depicted in Figure 3, VPR first translates structured visual information into parametric representations. Critically, such structured inputs are constrained to three primitive types: points, lines, and circles, which constitute the foundational elements of geometric constructions Trinh et al. (2024). For lines, parametric representations are derived by fitting straight-line models to the input data; for circles, parameters are extracted directly as center coordinates and radii. Subsequently, an optimal one-to-one correspondence $\mathcal{H}$ between predicted and ground-truth primitives is established via a multi-class Hungarian algorithm. This matching minimizes the total assignment cost $\mathcal{S}_{\mathrm{VPR}}$. The complete computational procedure is formalized in Algorithm 1.

## C.3  THE DETAILS OF INTLZR TRAINING

As shown in Table 10, the Knowledge Internalization Reward (IntlzR) model is trained on the MATHCOG-IntlzR subset containing 10k pairs (one positive and five negatives per pair). We initialize the reward model with Qwen2.5-VL-3B (Bai et al., 2025) using the HuggingFace Transformers library, and train it for 3 epochs with a batch size of 64 and a learning rate of $7 \times 10^{-6}$. It is worth noting that we adopt Softmax-DPO for training, with the objective function defined in Eq. 2, where the temperature parameter $\beta$ is fixed to 1.

IntlzR is implemented as a reward model trained using Softmax-DPO on contrastive trajectory pairs generated in the MATHCOG-IntlzR subset. For each positive trajectory, we use an LLM to generate five negative trajectories by injecting one of the five structured reasoning-error types into the "Thinking" part, and all synthetic negatives are manually verified. Each trajectory is encoded into a hidden representation $h$, and IntlzR learns a scalar score $R_{\mathrm{IntlzR}}(h) \in [0, 1]$ that reflects its consistency with the visual evidence. During training, IntlzR is optimized with the Softmax-DPO objective so that positive trajectories receive higher scores than their corresponding negative trajectories.

## C.4  THE DETAILS OF INFR

**Accuracy Reward.** The accuracy reward evaluates whether the response is correct. Following Shao et al. (2024), we assess correctness solely based on the `answer` field: the reward is 1 if the answer is correct and 0 otherwise.

Table 9: SFT configurations.

| Config | Setting |
|---|---|
| epochs | 1 |
| batch size | 64 |
| base learning rate | $1 \times 10^{-5}$ |
| optimizer | AdamW |
| LR scheduler | Cosine |
| weight decay | 0.01 |
| gradient clipping | 1.0 |
| memory optimization | ZeRO-2 |

Table 10: IntlzR configurations.

| Config | Setting |
|---|---|
| epochs | 3 |
| batch size | 64 |
| Base learning rate | $7 \times 10^{-6}$ |
| optimizer | AdamW |
| $\beta$ | 1 |
| memory optimization | ZeRO-2 |

Table 11: RL configurations.

| Config | Setting |
|---|---|
| epochs | 1 |
| batch size | 16 |
| base learning rate | $1 \times 10^{-6}$ |
| trajectory sampling | 8 |
| rollout generation | vLLM |
| precision | `bfloat16` |
| gradient accumulation | enabled |
| KL penalty coefficient | 0.001 |
| memory optimization | ZeRO-2 |

---

**Algorithm 1:** Multi-Class Matching for VPR

---

**Input:** Ground truth primitives $\mathcal{G} = \{G_1, G_2, \ldots, G_m\}$ with class labels
$\qquad \in \{\text{point}, \text{line}, \text{circle}\}$,
Predicted primitives $\mathcal{P} = \{P_1, P_2, \ldots, P_n\}$ with class labels $\in \{\text{point}, \text{line}, \text{circle}\}$
**Output:** Optimal matching $\mathcal{H} \subseteq \mathcal{G} \times \mathcal{P}$,
Total VPR cost $\mathcal{S}_{\text{VPR}} = \sum_{(i,j) \in \mathcal{H}} \|P_j - G_i\|_2$

$\mathcal{H} \leftarrow \emptyset; \quad \mathcal{S}_{\text{VPR}} \leftarrow 0$
$\mathcal{K} \leftarrow \{\text{point}, \text{line}, \text{circle}\}$                       `// Primitive classes`

**foreach** *class* $k \in \mathcal{K}$ **do**
    $\mathcal{G}_k \leftarrow \{G_i \in \mathcal{G} \mid \text{class}(G_i) = k\}$         `// GT primitives of class k`
    $\mathcal{P}_k \leftarrow \{P_j \in \mathcal{P} \mid \text{class}(P_j) = k\}$     `// Predicted primitives of class k`
    **if** $|\mathcal{G}_k| > 0$ **and** $|\mathcal{P}_k| > 0$ **then**
        `// Construct cost matrix` $C_k \in \mathbb{R}^{|\mathcal{G}_k| \times |\mathcal{P}_k|}$
        **for** $i \leftarrow 1$ **to** $|\mathcal{G}_k|$ **do**
            **for** $j \leftarrow 1$ **to** $|\mathcal{P}_k|$ **do**
                $C_k[i,j] \leftarrow \|\phi_k(G_k^{(i)}) - \phi_k(P_k^{(j)})\|_2$    `// L2 distance in parameter`
                `space`
            **end**
        **end**
        `// Apply Hungarian algorithm for optimal assignment`
        $\mathcal{H}_k \leftarrow \text{HUNGARIAN}(C_k)$      `// Returns matching pairs` $\mathcal{H}_k \subseteq \mathcal{G}_k \times \mathcal{P}_k$
        $\mathcal{S}_k \leftarrow \sum_{(i,j) \in \mathcal{H}_k} C_k[i,j]$       `// Class-specific cost` $\sum \|P_j - G_i\|_2$
        $\mathcal{H} \leftarrow \mathcal{H} \cup \mathcal{H}_k$
        $\mathcal{S}_{\text{VPR}} \leftarrow \mathcal{S}_{\text{VPR}} + \mathcal{S}_k$
    **end**
    **else if** $|\mathcal{G}_k| > 0$ **then**
        $\mathcal{S}_{\text{VPR}} \leftarrow \mathcal{S}_{\text{VPR}} + \lambda_{\text{FN}} \cdot |\mathcal{G}_k|$                `// Penalty for missed GT`
    **end**
    **else if** $|\mathcal{P}_k| > 0$ **then**
        $\mathcal{S}_{\text{VPR}} \leftarrow \mathcal{S}_{\text{VPR}} + \lambda_{\text{FP}} \cdot |\mathcal{P}_k|$             `// Penalty for false positives`
    **end**
**end**
**return** $\mathcal{H}$, $\mathcal{S}_{\text{VPR}}$

---

**Format Reward.** The format reward verifies whether the response adheres to the required output schema: the model must produce a JSON-style response, *i.e.*, `<WATCHING> ... </WATCHING> <THINKING> ... </THINKING> <ANSWER> ... </ANSWER>`. It returns 1 if the response is compliant and 0 otherwise.

Finally, we have the Inference Reward $R_{InfR}$:

$$R_{InfR} = R_{Acc} + R_{Fmt}. \tag{12}$$

## C.5 DETAILS OF SFT AND RL TRAINING

For supervised fine-tuning (SFT), we utilize the MATHCOG-SFT subset, which consists of 100,000 curated samples. Both the MATHCOG-SFT and MATHCOG-RL splits are organized into three components—`<WATCHING>`, `<THINKING>` and `<ANSWER>`. In our training pipeline, we first perform supervised fine-tuning (SFT) on the MATHCOG-SFT, where the model is trained on the "Watching" sequences together with the corresponding `<THINKING>` and`<ANSWER>` to strengthen its basic perceptual and reasoning abilities. The model is optimized for 1 epoch with a batch size of 64 and a learning rate of $1 \times 10^{-5}$. We adopt AdamW as the optimizer, using a cosine learning rate scheduler, a weight decay of 0.01, and gradient clipping at 1.0 to ensure stable updates.

For RL, the training is conducted on the MATHCOG-RL subset (10k samples) for 1 epoch, with a learning rate of $1 \times 10^{-6}$ and batch size of 16. For each input, we sample 8 candidate trajectories

using temperature-controlled decoding. The rewards are computed by combining Synergistic Visual Rewards (SynVRs), Knowledge Internalization Reward (IntlzR), and Inference Reward (InfR), with their weights fixed to $(1, 1, 1)$. We implement reinforcement learning under the VERL (Sheng et al., 2025) framework on 16 NVIDIA A100 GPUs, with distributed acceleration provided by vLLM. Mixed-precision training is conducted in `bfloat16`, and gradient accumulation is employed to simulate larger effective batch sizes. We adopt DeepSpeed ZeRO-2 for memory-efficient optimization, and the KL penalty coefficient is set to $0.001$ to stabilize policy updates.

## C.6 Details of the Reasoning Drift Precision Metric

As illustrated in Figure 1, we assess the precision of reasoning drift: whether each reasoning step is faithfully grounded in the perceived structure. Concretely, a generated solution is decomposed into steps $\{s_t\}_{t=1}^T$. From each step, we extract the referenced visual cues (primitives and relations) $\mathcal{R}_t$ by GPT-5 OpenAI (2025), and from the perception parse $y^{w*}$ we obtain the set of perceived elements $\mathcal{P}$. A step is deemed grounded if its visual claims are entailed by the perception, formalized by a consistency predicate, *i.e.*, $D(s_t) = 1$. The precision of reasoning drift detection for a response is given by

$$\text{Prec} = \frac{1}{T} \sum_{t=1}^T \mathbf{1}(D(s_t) = 1), \tag{13}$$

which measures the fraction of flagged steps that are truly ungrounded; higher values indicate fewer false positives and better alignment between perception and reasoning.

# D ADDITIONAL ANALYSIS

## D.1 ANALYSIS ON THE OVERALL EFFECTIVENESS OF COGFLOW

Figure 14 compares the performance of the base model (Qwen2.5-VL-7B), the model after SFT, and our full framework (COGFLOW). The base model achieves 46.30/40.10 on CoT-E/Acc, while SFT brings moderate improvements to 50.70/42.90 (+4.40/+2.80; approximately +9.5%/+7.0%). Building upon this, COGFLOW, trained with visual rewards and reinforcement learning, achieves substantial gains of 66.01/56.22, outperforming SFT by +15.31/+13.32 (around +30.2%/+31.0%) and the base model by +42.6%/+40.2%. Notably, CoT-E consistently surpasses Acc across all settings, with the largest margin observed in COGFLOW (9.79 points), indicating that our approach not only improves final answer accuracy but also significantly enhances the reliability and coherence of intermediate reasoning chains.

Finally, as shown in Table 12, COGFLOW attains an overall score of 76.8 (ALL), exceeding recent 7B-class multimodal reasoners and surpassing several general-purpose systems reported here (*e.g.*, GPT-4o 63.8, Claude-3.5-Sonnet 67.7, Gemini-2.0-Flash 73.4). While Doubao-pro-1.5 reaches a higher ALL (79.5), COGFLOW delivers the strongest category scores. The pronounced gains on geometry-centric GPS and text-heavy TQA align with our design: SynVRs stabilize fine-grained perceptual parsing, IntlzR enforces faithful internalization of visual cues, and VGPO optimizes reasoning under multi-signal feedback. Collectively, these components improve not only answer accuracy but also the consistency of "see correctly ⇒ internalize faithfully ⇒ reason coherently" across MathVista's diverse task types.

## D.2 MORE ANALYSIS ON SYNERGISTIC VISUAL REWARDS

We assess perception fidelity by comparing the model's structured output with the ground-truth primitives. Let $\mathcal{E}^{\text{pred}}$ and $\mathcal{E}^{\text{gt}}$ denote the predicted and gold sets of primitives (points/lines/circles). An LLM-based matcher sequentially aligns predicted elements to $\mathcal{E}^{\text{gt}}$ (type- and parameter-consistent), yielding a match set $\mathcal{M}$. We compute

$$\text{Precision} = \frac{|\mathcal{M}|}{|\mathcal{E}^{\text{pred}}|}, \qquad \text{Recall} = \frac{|\mathcal{M}|}{|\mathcal{E}^{\text{gt}}|}, \qquad \text{F1} = \frac{2\,\text{Precision} \cdot \text{Recall}}{\text{Precision} + \text{Recall}}. \qquad (14)$$

Figure 15 shows the F1 distribution under three settings (w/o VSR, w/o VPR, w/ VSR): removing either reward lowers perception fidelity, while enabling **VSR** produces the right-most, highest-centered distribution, indicating more primitives are correctly grounded; **VPR** provides complementary but weaker regularization.

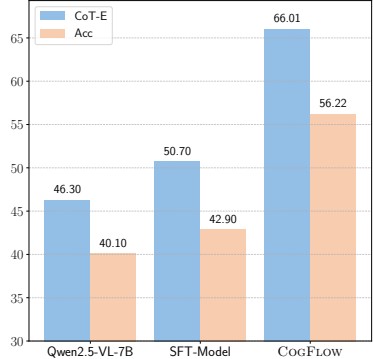

Figure 14: **Effectiveness of COGFLOW.** We compare the base model (Qwen2.5-VL-7B) and its SFT-Model against COGFLOW.



Figure 15: **Perception F1 of SynVRs variants.** The W/SynVRs achieves the best performance.

Table 12: Accuracy (%) results on MathVista dataset.

| Method | All | FQA | GPS | MWP | TQA | VQA |
|---|---|---|---|---|---|---|
| Gemini-1.0-Pro (Team et al., 2025) | 47.7 | 48.3 | 35.1 | 50.5 | 65.8 | 42.5 |
| GPT-4V (OpenAI, 2023) | 49.9 | 43.1 | 50.5 | 57.5 | 65.2 | 38.0 |
| GPT-4o (OpenAI, 2024) | 63.8 | - | - | - | - | - |
| Claude 3.5 Sonnet (Anthropic, 2024) | 67.7 | - | - | - | - | - |
| Gemini-2.0-Flash (Team et al., 2025) | 73.4 | - | - | - | - | - |
| Doubao-pro-1.5 (Seed et al., 2025) | **79.5** | **77.7** | **88.9** | **86.0** | **82.3** | **62.0** |
| LLaVA-1.5-7B (Liu et al., 2024b) | 20.0 | 22.7 | 7.7 | 11.8 | 26.6 | 33.0 |
| LLaMA-Adapter-V2-7B (Gao et al., 2023) | 23.9 | 21.2 | 25.5 | 11.3 | 32.3 | 31.8 |
| TinyLLaVA-3B (Zhou et al., 2024) | 26.7 | 20.4 | 19.7 | 23.1 | 44.9 | 31.8 |
| LLaVA-1.5-13B (Liu et al., 2024b) | 26.8 | 19.7 | 20.2 | 18.8 | 45.6 | 36.9 |
| mPLUG-Owl2-7B (Ye et al., 2024) | 22.2 | 22.7 | 23.6 | 10.2 | 27.2 | 27.9 |
| MiniGPT-v2-7B (Chen et al., 2023a) | 23.1 | 18.6 | 26.0 | 13.4 | 30.4 | 30.2 |
| G-LLaVA-7B (Gao et al., 2025) | 25.1 | 19.1 | 48.7 | 3.6 | 25.0 | 28.7 |
| VCAR (Jia et al., 2024b) | 33.7 | 30.9 | 34.6 | 38.7 | 37.3 | 28.5 |
| SPHINX-Plus (Liu et al., 2024a) | 36.7 | 54.6 | 16.4 | 23.1 | 41.8 | 43.0 |
| SVE-Math-7B (Zhang et al., 2025b) | 37.4 | 31.9 | 53.9 | 29.0 | 41.4 | 30.8 |
| MultiMath-7B (Peng et al., 2024a) | 50.0 | 40.1 | 66.8 | 61.8 | 50.0 | 33.0 |
| X-REASONER (Liu et al., 2025) | 69.0 | - | - | - | - | - |
| VL-Rethinker-7B (Wang et al., 2025a) | 73.7 | - | - | - | - | - |
| ReVisual-R1 (Chen et al., 2025c) | 73.1 | - | - | - | - | - |
| WeThink (Yang et al., 2025) | 70.9 | - | - | - | - | - |
| Skywork-R1V-38B (Wang et al., 2025c) | 60.6 | - | - | - | - | - |
| CogFlow-7B | 76.8 | 70.4 | 93.1 | 73.7 | 86.9 | 59.3 |

Table 13: **Ablation analysis of visual rewards and RL strategies.** We compare GRPO with PPO and DAPO under consistent reward settings.

| Method | Configuration | FlowVerse CoT-E(%) | FlowVerse Acc (%) | Training Time (h) |
|---|---|---|---|---|
| DAPO Yu et al. (2025) | w/o SynVRs, w/o IntlzR | 56.1 | 45.8 | 4.1 |
| | w SynVRs, w/o IntlzR | 59.7 | 50.7 | 4.6 |
| | w SynVRs, w IntlzR | 62.8 | 54.1 | 4.6 |
| PPO Yu et al. (2022) | w/o SynVRs, w/o IntlzR | 55.9 | 44.6 | **3.2** |
| | w SynVRs, w/o IntlzR | 58.2 | 48.3 | 3.7 |
| | w SynVRs, w IntlzR | 61.5 | 53.9 | 3.7 |
| GRPO Shao et al. (2024) | w/o SynVRs, w/o IntlzR | 56.7 | 47.6 | 4.2 |
| | w SynVRs, w/o IntlzR | 61.4 | 52.9 | 4.7 |
| | w SynVRs, w IntlzR | 64.4 | 55.1 | 4.8 |
| VGPO (ours) | w/o SynVRs, w/o IntlzR | 57.4 | 48.7 | 4.6 |
| | w SynVRs, w/o IntlzR | 63.2 | 54.7 | 5.3 |
| | w SynVRs, w IntlzR | **66.0** | **56.2** | 6.1 |

## D.3 THE IMPACT AND EFFICIENCY OF THE VISION GATE

**The Impact of the Visual Gate in the Training Phase.** To assess how the visual gate improves perceptual quality during VGPO training, we analyze the visual-reward scores produced by its conditional re-generation procedure. Figure 16 reports three visual rewards score distributions, each computed over the full training set but corresponding to different re-generation under visual gate with the threshold $\tau$: (i) the first-attempt scores for all instances; (ii) the post-gating scores after one re-generation round, where only instances with $S_{vis} < \tau$ are re-generated and use their second-attempt scores while the rest keep their first-attempt scores; and (iii) the post-gating scores after a second re-generation round, where instances still below $\tau$ after the second attempt are re-generated again and use their third-attempt scores, while all others keep the earliest score at which they pass the gate. We visualize these distributions together with the threshold $\tau$ and compute the corresponding pass rates as the fraction of examples whose score exceeds $\tau$ at each stage. The results show

Table 14: **Analysis of Inference Schemes**. We evaluate CogFlow and the VLM-R1 baseline under these three settings from the FlowVerse with $k = 3$.

| Model | FlowVerse CoT-E | Acc | Inference Time (h / 1000 samples) |
|---|---|---|---|
| VLM-R1-7B (single-pass) | 56.17 | 47.63 | 2.77 |
| VLM-R1-7B (best-of-3 full) | 59.45 | 49.59 | 5.54 |
| CogFlow-7B (single-pass) | 64.51 | 55.22 | 2.72 |
| CogFlow-7B (best-of-3 full) | 66.04 | 56.17 | 5.52 |
| **CogFlow-7B (visual gate)** | **66.03** | **56.24** | **3.06** |

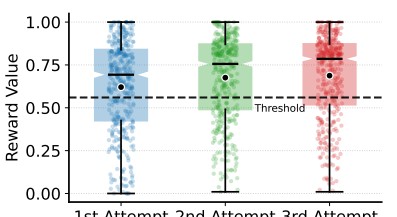

Figure 16: **The distribution of visual reward scores during training.** Attempts with scores above the "Threshold" are accepted.

that the score mass shifts upward and becomes increasingly concentrated above $\tau$, indicating that the visual gate acts as an effective perceptual quality-control mechanism that improves perceptual grounding rather than arbitrarily resampling trajectories.

Meanwhile, as evidenced by Table 13, enabling VGPO with the visual gate incurs only a modest overhead of $+1.3$ training hours, which is acceptable for standard MLLM post-training.

**The Impact of the Visual Gate in the Inference Phase.** Moreover, we add additional experiments to further analyze the impact of the visual gate at inference. Specifically, we compare three inference schemes:

- **Single-pass:** The model generates a full trajectory (`<WATCHING>`, `<THINKING>`, `<ANSWER>`), which is identical to standard MLLM inference.

- **Best-of-$k$ full:** The model generates $k$ full trajectories and selects the one whose final answer matches the ground truth. We note that this is not a realistic inference strategy, as it relies on access to ground-truth answers.

- **Visual gate (best-of-$k$ perception):** CogFlow samples $k$ alternative perception trajectories, selects the one with the highest visual score, and then performs a single reasoning pass conditioned on the selected perception.

As shown in Table 14, to enable a fair and controlled comparison, we evaluate CogFlow and an open-source R1-style baseline (VLM-R1) under all three inference schemes with a fixed $k = 3$. We additionally report the *average time required to process 1,000 examples* (computed by selecting 1,000 samples and averaging the end-to-end inference time) for each setting.

The results show that CogFlow outperforms VLM-R1 under *both* the single-pass and best-of-3 full settings, indicating that the observed gains are not attributable to more aggressive sampling. Moreover, the visual gate (best-of-3 perception) achieves nearly the same accuracy as best-of-3 full responses while requiring substantially less inference time (2.72 h vs. 3.06 h vs. 5.52 h). Overall, these results support that our comparison is fair, and that CogFlow's visual-gated inference provides a more favorable computation–performance trade-off than naive best-of-$k$ sampling over full responses.

**The Impact of $k$ in Visual Gate** Finally, we ablate the perception sampling number $k$ used by the visual gate at *both* training time and inference time. Table 15 varies the training-time $k$ while keeping the inference-time gate fixed to $k = 3$, whereas Table 16 varies the inference-time $k$ with the training-time gate fixed to $k = 3$. Across both ablations, $k = 3$ consistently yields a favorable performance–computation trade-off, achieving most of the attainable accuracy gains without incurring excessive additional cost. Accordingly, we adopt $k = 3$ as the default setting for both stages.

## D.4 Analysis on Visual-Gated Policy Optimization

Table 13 presents an ablation analysis across different reinforcement learning algorithms (DAPO (Yu et al., 2025), PPO (Schulman et al., 2017), GRPO (Shao et al., 2024), and our VGPO) under consis-

Table 15: **Analysis of perception sample number $k$ in the visual gate during training.** Inference-time visual gate fixed to $k = 3$.

| $k$ | FlowVerse | | Avg Training Time (h) |
| --- | --- | --- | --- |
| | CoT-E | Acc | |
| 1 | 64.40 | 55.13 | 4.85 |
| 3 | 66.03 | 56.24 | 6.17 |
| 5 | 66.60 | 56.52 | 7.91 |

Table 16: **Analysis of perception sample number $k$ in the visual gate during inference.** Training-time visual gate fixed to $k = 3$.

| $k$ | FlowVerse | | Avg Inference Time (h / 1000 samples) |
| --- | --- | --- | --- |
| | CoT-E | Acc | |
| 1 | 64.51 | 55.22 | 2.72 |
| 3 | 66.03 | 56.24 | 3.06 |
| 5 | 66.36 | 56.55 | 3.71 |

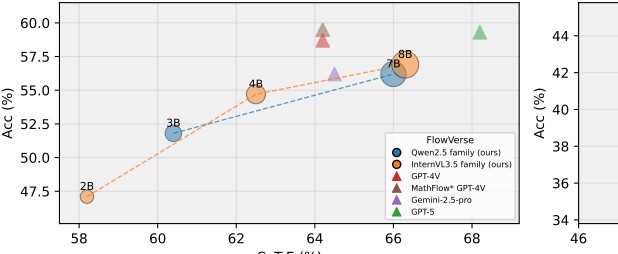
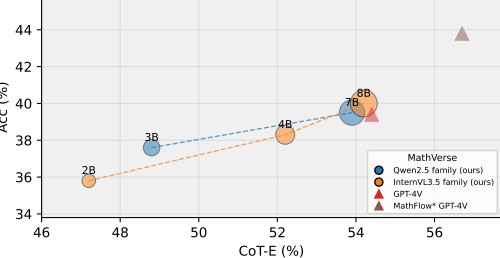

Figure 17: Scalability and broader applicability of COGFLOW.

tent reward configurations based on the MATHCOG. Several clear trends emerge. First, the inclusion of visual perceptual reward (VPR) and Knowledge Internalization Reward (IntlzR) consistently improves performance across all methods, confirming their complementary roles in strengthening visual grounding. Second, GRPO outperforms both PPO and DAPO in all settings, highlighting its stability and effectiveness in geometry-oriented reasoning tasks. Most importantly, our proposed VGPO substantially surpasses all baselines, achieving 66.0%/56.2% accuracy on FlowVerse and MathVerse, respectively. This represents an improvement of over +6 points compared to GRPO with VPR and IntlzR, demonstrating the advantage of explicitly disentangling perception and reasoning optimization. Although VGPO incurs slightly higher training costs, the performance gain justifies the trade-off.

### D.5 CLARIFYING THE COGNITIVE-SCIENCE ANALOGY

First, the "watching" stage in COGFLOW is not intended to correspond to internalization. Its function is limited to extracting low-level structured visual information from an image—such as points, lines, circles, spatial groupings, and relative configurations. These representations are still purely perceptual: they serve as a normalized, noise-reduced description of the raw visual scene, but they do not yet constitute conceptual understanding or task-oriented reasoning.

The subsequent internalization stage is distinct and is designed precisely to bridge perception and reasoning. It transforms the structured primitives produced in the watching stage into a conceptually meaningful representation, integrating visual evidence with symbolic relations and problem-specific abstractions. Internalization therefore plays the role of aligning perceptual inputs with the evolving reasoning state. For instance: Type 1 (Omit or misbind primitives) errors stem from incorrect or incomplete internalization of otherwise correctly perceived primitives. They reflect failures in mapping the structured perceptual tokens into a coherent conceptual state—hence they lie at the interface between perception and reasoning. Type 4 (Invoke external theorems inappropriately) and Type 5 (Refer inconsistently to established elements) arise when the internalized conceptual representation is misinterpreted, misapplied, or inconsistently used within multi-step reasoning. These error types are characteristic of reasoning drift under an inadequately grounded internal state.

### D.6 DEEPER INSIGHTS INTO THE MECHANISM OF MULTI-STAGE REWARD STRUCTURE

On the one hand, each reward is designed to target a distinct failure mode in visual mathematical reasoning: Firstly, the Synergistic Visual Rewards (SynVRs), which comprise VPR and VSR, are designed to address perceptual failures. Figure 5 presents an ablation of the SynVRs components. The results show that adding either VPR or VSR alone consistently improves performance, and the best results are obtained when both are enabled. Figure 15 further reports the perception F1 scores

for different SynVRs variants, indicating that removing either reward degrades perception fidelity. Finally, the error-type analysis in Figure 9 demonstrates that SynVRs effectively reduce perception-related errors, thereby validating their effectiveness. Taken together, these results show that VPR and VSR provide complementary supervision signals: VPR enforces local geometric fidelity, while VSR encourages global perceptual coherence in terms of overall style and layout, and together they form trustworthy visual cues that serve as a robust foundation for effective visual mathematical reasoning, which is consistent with the conclusions reported in prior studies (Guo et al., 2025c; Chen et al., 2025a). Secondly, the Knowledge Internalization Reward (IntlzR) is designed to alleviate reasoning drift by bridging the perception and reasoning stages, encouraging the model to produce structured, reasoning-ready outputs (*i.e.*, knowledge-internalized representations (Ryan & Connell, 1989)) that provide a more reliable foundation for subsequent reasoning. The ablation study on IntlzR (Figure 6) shows that removing any single error type consistently degrades performance, indicating that each error type provides complementary supervision. The largest drops occur when excluding the omission/misbinding primitives or contradicting geometric constraints error types, highlighting that correctly binding primitives and respecting core geometric constraints are most critical for keeping reasoning tied to perception. In addition, the error-type analysis in Figure 9 further demonstrates that IntlzR effectively reduces Knowledge Internalization Errors, which in turn leads to improved overall performance. Thirdly, the Inference Reward (InfR) ensures task-level correctness and proper output structure at the final reasoning stage (Shao et al., 2024).

On the other hand, this multi-stage reward structure provides a principled solution to the credit-assignment problem: instead of relying on a single sparse task-level reward, the stage-wise signals supply informative gradients at different points of the perception–internalization–reasoning pipeline. The ablation on COGFLOW's components (Table 5) shows that both SynVRs and IntlzR are essential for achieving strong performance, and that the best results are obtained when all rewards are enabled. In addition, the reasoning drift analysis across three representative pipelines (Figure 1b) demonstrates that COGFLOW's reasoning trajectories are more stable and better aligned with the visual information, which is consistent with the observed performance gains. Moreover, the error-type analysis in Figure 9 shows that, compared with the Baseline+VGPO setting, COGFLOW substantially reduces Perception Errors, knowledge-internalization errors, and Reasoning Errors. This indicates that the three rewards together form a coherent cognitive paradigm: improved perception supports more reliable internalization, and IntlzR further aligns the reasoning trajectory with the visual evidence

### D.7 SIZES AND ARCHITECTURES ABLATION.

To demonstrate the scalability and broader applicability of COGFLOW, we now explicitly evaluate COGFLOW across multiple backbone architectures and parameter scales, rather than only on Qwen2.5-VL-7B (see Figure 17). Specifically, we report results for Qwen2.5-VL (3B and 7B) and InternVL3.5 (2B, 4B, and 8B). The results show that: (1) COGFLOW consistently outperforms models of comparable size, and even with only 2B or 4B parameters it achieves competitive performance; (2) the performance gains are maintained or even amplified as model capacity increases; (3) the improvements hold for both the Qwen and InternVL families, indicating that the framework is not tied to a specific architecture. These findings provide empirical evidence that COGFLOW is scalable and broadly applicable beyond a single model configuration.

### D.8 SOURCE OF PERFORMANCE IMPROVEMENT

Since MATHCOG is explicitly designed to support the training of COGFLOW, the method and the dataset are inherently coupled and cannot be treated as fully independent components. Nevertheless, for completeness, we conduct a controlled analysis to further investigate to what extent the observed performance gains arise from the dataset itself versus from the proposed training framework. We conducted an ablation study in the Table 13 where we fix the training data to MATHCOG and vary only the post-training algorithm. Starting from the same base model, we compare standard PPO, DAPO and our proposed VGPO. The results show that all methods benefit from training on MATHCOG, indicating that VGPO consistently achieves the best performance among all post-training methods under the same data. This suggests that the performance improvements cannot be attributed to the dataset alone, and that the COGFLOW framework and its multi-stage optimization play a central role in the observed gain.

Table 17: **Comparison of pre-trained models on FlowVerse.** Results show that using FG-CLIP with a ViT-L-14 backbone yields the best performance.

| MetaCLIP2 | | | | FG-CLIP | | | |
|---|---|---|---|---|---|---|---|
| ViT-B-16 | | ViT-L-14 | | ViT-B-16 | | ViT-L-14 | |
| CoT-E | Acc | CoT-E | Acc | CoT-E | Acc | CoT-E | Acc |
| 62.5 | 53.9 | 63.2 | 54.5 | 63.3 | 54.6 | **63.8** | **55.3** |

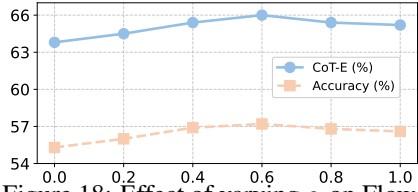

Figure 18: Effect of varying $\alpha$ on Flow-Verse performance.

### D.9    ABLATIONS ON INDIVIDUAL SYNVRS COMPONENTS

We note that the Figure 5 already included ablations of each SynVRs component. However, we did not previously investigate how different mixing ratios between VPR and VSR affect performance.

We therefore conduct an ablation study on the FlowVerse dataset (see Figure 18) by varying the weighting coefficient $\alpha$ while keeping IntlzR and the visual gate enabled. The results show that performance peaks at $\alpha = 0.6$, indicating that balancing VPR and VSR yields the most effective supervisory signal. Moreover, the fact that $\alpha = 0$ underperforms $\alpha = 1$ suggests that parameter-level supervision (VPR) contributes more substantially to perceptual fidelity than semantic similarity alone (VSR).

### D.10    EVALUATING FG-CLIP FOR SIMILARITY REWARD

We further evaluate the FG-CLIP similarity module by replacing it with several alternative pre-trained models (*e.g.*, MetaCLIP2 (Chuang et al., 2025)) of comparable capacity. Across these variants, the FG-CLIP (ViT-L-14) configuration consistently achieves the best overall performance on FlowVerse, indicating that FG-CLIP provides a stronger and more discriminative supervision signal for our VSR component than the alternative encoders

### D.11    FURTHER ANALYSIS OF ERROR TYPES

From Figure 9, we observe that the *Baseline+SynVRs* setting leads to only a modest reduction in perception-related errors, but a more substantial reduction in reasoning-related errors (see also Figure 9 in the original submission). We explain this phenomenon as follows.

First, we view this as a natural consequence of the fact that perception, internalization, and reasoning are not independent modules: improvements in perception inevitably propagate to, and influence, the subsequent reasoning process. We conduct a systematic case study to analyze the error compensation mechanism. In the illustrated example, the perception stage exhibits partial errors: the coordinates of point $A$ are misidentified, which in turn corrupts lines $AB$ and $AD$. Nevertheless, during the subsequent reasoning stage, the model still produces the correct answer by accurately perceiving other visual primitives (*e.g.*, points $B$ and $C$) and performing precise internalization. This analysis reveals a cognitive processing pattern in which response generation proceeds sequentially through perception $\Rightarrow$ internalization $\Rightarrow$ reasoning, and suggests that strengthened perception and internalization capabilities can mitigate Reasoning Errors even in the presence of localized perception failures.

In addition, among the Baseline+SynVRs, Baseline+IntlzR, and Baseline+VGPO configurations, Baseline+SynVRs already yields the largest reduction in Perception Errors, further demonstrating that SynVRs provide effective improvements to the perceptual component.

### D.12    CASE STUDIES

As shown in Figure 19, consider a circle with center $O$ and points $A, B, C$ on the circle, where $\angle ABO = 50°$. A typical failure mode is primitive misbinding: one baseline incorrectly treats the given condition as $\angle AOB = 50°$ and then applies the inscribed-angle theorem to obtain $\angle ACB = 25°$. Another baseline perceives the condition correctly but makes an invalid geometric inference by asserting $\angle AOB = 2\angle ABO$, which yields $\angle ACB = 50°$. In contrast, with internalized primitives and visual rewards (VPR/IntlzR), our model preserves consistent grounding. Since $OA = OB$,

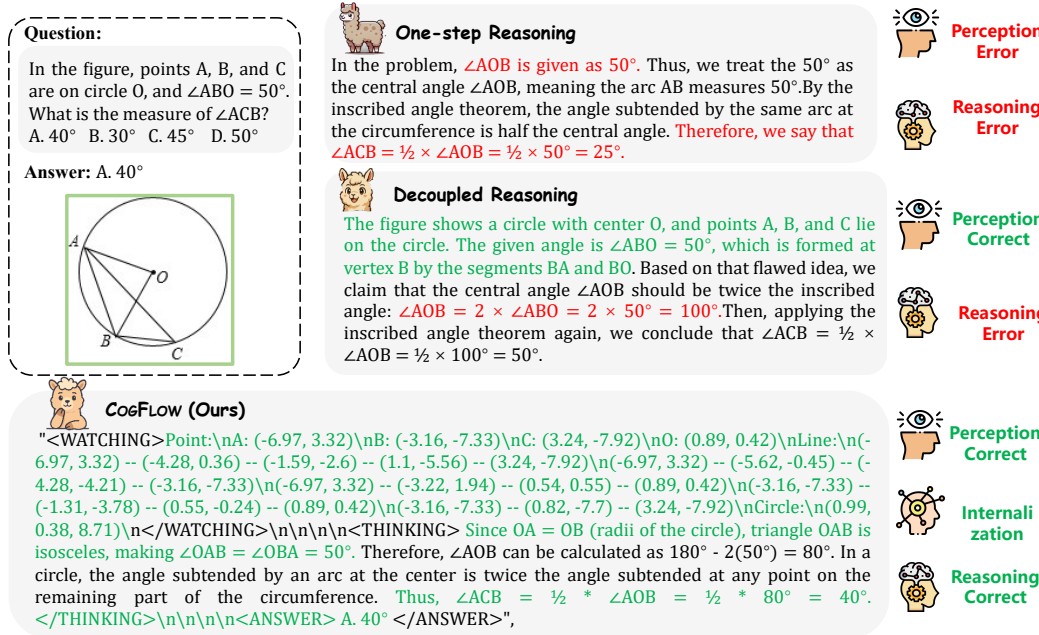

Figure 19: An example of the solution generated by COGFLOW.

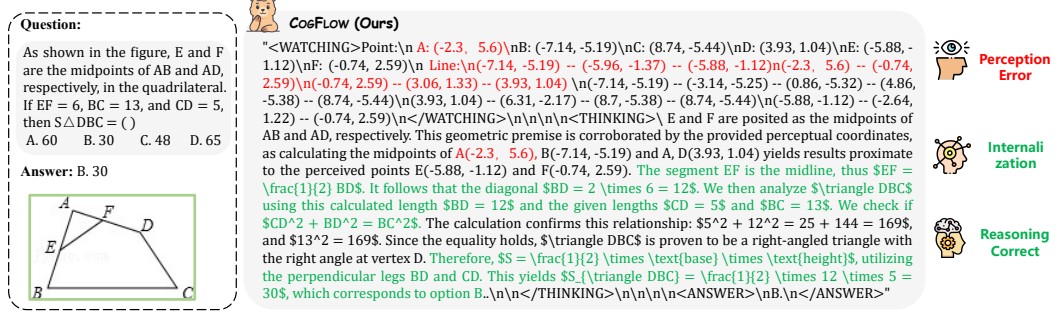

Figure 20: Example solution generated by COGFLOW under partially erroneous perception.

$\triangle OAB$ is isosceles, so $\angle OAB = \angle ABO = 50°$. It follows that $\angle AOB = 180° - 2\angle OAB = 80°$, and by the inscribed-angle theorem $\angle ACB = \frac{1}{2}\angle AOB = 40°$, which is correct. This example illustrates that disentangling perception from reasoning and supervising both stages with VPR/IntlzR mitigates primitive misbinding and discourages spurious theorem application.

### D.13 ANALYSIS OF ERROR COMPENSATION MECHANISM

As shown in Figure 20, we conduct a systematic case study to analyze the error compensation mechanism. In this instance, perception contains partial errors: the coordinates of point $A$ are misidentified, consequently corrupting lines $AB$ and $AD$. Crucially, during subsequent reasoning, the model produces correct answers by accurately perceiving other visual primitives (*e.g.*, points $B$ and $C$) and performing precise *knowledge internalization*. Our analysis reveals a cognitive paradigm where response generation progresses sequentially through perception⇒internalization⇒reasoning. This demonstrates that enhanced perception and internalization capabilities can mitigate Reasoning Errors even when partial perception failures occur.

