# OpenReview forum: "CogFlow: Bridging Perception and Reasoning through Knowledge Internalization for Visual Mathematical Problem Solving"
_ICLR.cc/2026/Conference — ICLR 2026 Poster_

### Official Review · Reviewer_Wrhz · 2025-10-31

**Soundness:** 3
**Presentation:** 3
**Contribution:** 3
**Rating:** 6
**Confidence:** 4

**Summary:**

The authors propose CogFlow, a VLM framework that represents images through geometric primitives before performing reasoning, and introduce MathCog, a new dataset annotated with these primitive-level representations. In CogFlow, visual primitives (e.g., points, lines, circles) are learned to be both geometrically accurate (correct coordinates) and semantically consistent with the original image. During training, primitive predictions with low visual score are regenerated using a thresholding mechanism, termed VGPO. At inference time, only similarity-based supervision is applied since ground-truth primitives are unavailable. To further enhance reasoning accuracy, the authors train a logical error reward that identifies inconsistencies in reasoning steps, which is then used to fine-tune the model to improve consistency.

**Strengths:**

1. **Strong results.** Achieves significant performance gains over comparable open-source baselines, showing consistent improvements across multiple benchmarks. Notably, the proposed method narrows the gap with, and in some cases surpasses, the performance of leading closed-source models.
2. **Clear presentation.** The writing is clear and organized, with helpful figures. Despite the many nuances of the proposed approach, the motivation and details are clearly presented (though some additional details may be missing, see weaknesses).
3. **Substantive contributions.** Introducing a new reasoning framework around improving VLM perception (CogFlow) and a complementary dataset (MathCog) to support the framework. This method and dataset, which results in strong perfomance, is a valuable addition to the field.

**Weaknesses:**

1. **Insufficient dataset details and transparency.** The description of the MathCog dataset (in the appendix) lacks important specifics about the data collection and annotation process. It remains unclear exactly what tasks human annotators performed, the scale of human effort involved, and the associated cost or time investment. Additionally, the process for generating negative samples should be more explicitly explained (e.g. prompts used).
2. **Ambiguity in source of performance improvements.**
It is difficult to disentangle the contribution of the CogFlow framework from that of the MathCog dataset. Since CogFlow is trained using MathCog, the performance gains might partially stem from the dataset's scope rather than the model's method. A useful comparison would be to evaluate other methods when trained on MathCog to better understand how much improvement arises from the dataset versus the proposed architecture. While Table 4 (with Table 1/2) gives some sense of this improvement, it is unclear how much improvement other methods would achieve given the same dataset.
3. **Misalignment with cognitive science framing.**
While the paper draws inspiration from cognitive science concepts, the mapping between these ideas and the proposed components is not entirely convincing. For instance, the "watching" tokens more closely resemble internalization (structured encoding of visual input) than perception. Similarly, the Visual Assessment Reward (VAR) appears to operate across multiple stages of cognition, including perception (error type 1) and reasoning (error types 3 and 4), rather than targeting a specific cognitive process. This partial misalignment makes the method more confusing, though the link can still provide good motivation.
4. **Missing experiments.** Missing comparison with SophiaVL-R1 [1], which may have stronger performance. Missing ablations for individual SVR components (VPR and VSR) (or alpha hyperparameter L248), as well as evaluations of how good FG-CLIP is for similarity reward.

**Suggestions:**

1. Spelling (L26 "grated"->"gated", L27 "that produce appear"->"that appear"). Fig 3 "Visual Similarity Reward" -> "Visual Semantic Reward" and Visual Parameter Reward -> parameterized.
2. Lots of acronyms (SVR, VAR, VGPO, VPR, VSR) makes method a bit confusing, and some acronyms aren't very descriptive (e.g. synergistic visual reward).
3. Equation 4 does not follow the algorithm as described. As described (L292), k=2,3 is not evaluated if k=1 passes the threshold, but equation 4 implies 2 and 3 are generated and checked (maximized).
4. Equation 3 should include preferred trajectory (s+) in the denominator sum, otherwise it's not softmax. Also Softmax-DPO should be better cited L272 [2]
5. Figure 2 is unclear whether the supervised fine-tuning (SFT) stage uses the "watching" and new "thinking" annotations, as the yellow/green outline suggests watching/thinking are only used during RL.
6. Expand one-sentence figure/table captions.

[1] Fan, Kaixuan, et al. "SophiaVL-R1: Reinforcing MLLMs Reasoning with Thinking Reward." arXiv preprint arXiv:2505.17018 (2025).

[2] Chen, Yuxin, et al. "On softmax direct preference optimization for recommendation." Advances in Neural Information Processing Systems 37 (2024): 27463-27489.

**Questions:**

1. In Figure 7, why does the Baseline+SVRs setting show limited reduction in perception errors, with most gains coming from reducing reasoning errors?
2. In Table 4, do all evaluations use the visual gate during inference? What happens if the model is trained without VGPO but still uses the gate at inference?
Do all evaluations in Table 4 use visual gate at inference? And what is the impact of training without VGPO but using gate at inference?
3. In Figure 2, what do the easy/medium/hard categories refer to?

---

> ### Author Response · Authors · 2025-11-27
> **Rebuttal by Authors [1/8]**
>
> We sincerely thank Reviewer `Wrhz` for recognizing our **strong results**, **clear presentation** and **substantive contributions**. In the following, we address the remaining concerns point by point.
>
> ## **Weakness1: Insufficient dataset details and transparency.**
> We apologize for the insufficient details and lack of clarity in our original description (Appendix C.2) of the MathCog dataset. We now provide a more detailed account of the data collection and annotation process.
>
> **Data collection.**
> To construct MathCog, we first collect a large pool of visual math problems from existing corpora, including MAVIS[1], Geo170K[2], and LLaVA-CoT[3]. We then perform a careful filtering process to obtain a high-quality subset of 111,752 problems. Specifically, we remove (1) problems whose visual content is dominated by charts with weak geometric relevance (e.g., histograms, pie charts) or artistic illustrations that deviate from geometric reasoning, and (2) diagrams whose shorter side is below 64 pixels or exceeds 2048 pixels, which would otherwise compromise the consistency and stability of model training
>
> **Visual Primitive Annotation.**
> We utilize the OpenCV library to detect pixel-coordinate-based visual primitives in the images from the collected data, including endpoints and intersections of line segments, as well as circular features. For circles, we directly extract center coordinates $(x, y)$ and radius $r$ in pixel space. For points (e.g., vertices and character annotations), we similarly obtain precise pixel coordinates and establish optimal correspondences between detected visual elements and character labels via the Hungarian algorithm, thereby obtaining final mappings such as $A:(x_1,y_1)$ and $B:(x_2,y_2)$. These outputs then undergo systematic validation by human annotators, who rectify missing primitives and geometric misalignments.
>
> **Reasoning Annotation.** Based on these visual primitives, we further construct reasoning trajectories from the solution rationales in the collected data. The goal is to (i) explicitly internalize the structured primitives into a symbolic representation, and (ii) perform step-by-step logical inference over this internalized state to answer the question. These trajectories are first generated with the assistance of GPT-5 and are subsequently verified and refined by human annotators. The prompt template used for this refinement is shown below:
> | Role | Content |
> |------|-------|
> | **System** | *# Your Role: expert math geometry teacher* |
> | **Objective** | You will be provided with a visual mathematics problem, along with its perception output `<Watching>` and a raw solution `<Thinking>`. Your task is to refine the solution such that the reasoning process explicitly internalizes the perceptual information before carrying out logical inference. |
> | **Output Format** | The output format should strictly follow the example:<br>-- Question: XXX <br>-- `<Watching>`: XXX <br>-- `<Thinking>`: XXX <br>-- **Refined `<Thinking>`:** XXX |
> | **Data to Refine** | -- Question: {question} <br>-- `<Watching>`: {watching} <br>-- `<Thinking>` (raw): {solution} <br>-- **Refined `<Thinking>`:** |
>
> **MathCog-VAR Annotation.**
> Building on MathCog-SFT, we construct MathCog-VAR as a set of contrastive reasoning trajectories with explicit perturbations:
> 1.  We first sample 10  ,000 examples as positive trajectories from the MathCog-SFT split.
> 2. We categorize reasoning errors into five types: (1) omitting or misbinding primitives, (2) introducing nonexistent facts, (3) contradicting geometric constraints, (4) invoking external theorems inappropriately, and (5) referring inconsistently to established elements. For each positive trajectory, we use LLM to generate negative trajectories conditioned on the positive one by explicitly instructing it to modify the <Thinking> part so that it exhibits a specific type of error while keeping the remaining reasoning as faithful as possible.

---

> ### Author Response · Authors · 2025-11-27
> **Rebuttal by Authors [2/8]**
>
> | **Error Type**                                | **Instruction**                                                                                                                                                                                                                                                                             | **Positive Data** |
> |-----------------------------------------------|---------------------------------------------------------------------------------------------------------------------------------------------------------------------------------------------------------------------------------------------------------------------------------------------|-------------------|
> | Omit or Misbind Primitives                    | *Starting from the correct reasoning, deliberately omit or misbind at least one basic geometric primitive (points, lines, circles, etc.), while keeping the overall reasoning fluent and locally plausible.*                                                                               | {positive data}   |
> | Introduce Nonexistent Facts                   | *Starting from the correct reasoning, introduce at least one geometric or numerical fact that is not supported by the given figure or problem statement, but make the reasoning appear locally coherent and natural.*                                                                       | {positive data}   |
> | Contradict Geometric Constraints              | *Starting from the correct reasoning, modify the chain of thought so that at least one step violates the true geometric constraints (e.g., equal lengths, parallelism, angle measures), while preserving a seemingly reasonable narrative.*                                                 | {positive data}   |
> | Invoke External Theorems Inappropriately      | *Starting from the correct reasoning, inappropriately invoke at least one external theorem or formula whose preconditions are not satisfied, or whose use is not justified by the internalized structure, while keeping the explanation linguistically smooth.*                            | {positive data}   |
> | Refer Inconsistently to Established Elements  | *Starting from the correct reasoning, alter the chain of thought so that references to previously established elements (points, lines, or relationships) become inconsistent, such as swapping labels or changing properties across steps, but without breaking the overall fluency of the text.* | {positive data}   |
>
>
> 3. We then conduct a dedicated human verification stage, during which annotators manually review and correct all trajectories that do not meet the required criteria. This process yields a final dataset of 10{,}000 positive trajectories and 50,000 carefully curated negative trajectories. A dedicated team of 31 professional annotators carried out this quality-assurance protocol over a one-month period, ensuring high-fidelity and consistent ground-truth annotations.
>
> We have provided a detailed explanation of the data collection and annotation process in **Appendix C.2**.
>
> [1] Zhang R, Wei X, Jiang D, et al. Mavis: Mathematical visual instruction tuning[J]. arXiv e-prints, 2024: arXiv: 2407.08739.
>
> [2] Gao Y, Wei X, Liu M, et al. Geo170k: A large-scale benchmark for geometry problem solving[J]. arXiv e-prints, 2023: arXiv: 2306.14216.
>
> [3] Xu Y, Wang P, Shao Z, et al. LLaVA-CoT: Let vision language models see the diagrams in visual math problems[J]. arXiv preprint arXiv:2503.16549, 2025.

---

> ### Author Response · Authors · 2025-11-27
> **Rebuttal by Authors [3/8]**
>
> ## **Weakness2: Ambiguity in source of performance improvements.**
> We acknowledge that disentangling the performance gains attributable to the CogFlow framework from those arising from the MathCog dataset is an important concern. While our original submission already addressed this point in Appendix E.2 in the initial submission, we have further expanded the analysis in the revised manuscript to more clearly explanation.
>
>
> Since MathCog is explicitly designed to support the training of CogFlow, the method and the dataset are inherently coupled and cannot be treated as fully independent components. **Nevertheless, for completeness, we conduct a controlled analysis to further investigate to what extent the observed performance gains arise from the dataset itself versus from the proposed training framework.**
> We conducted an ablation study in the table where we fix the training data to MathCog and vary only the post-training algorithm. Starting from the same base model, we compare standard PPO, DAPO and our proposed VGPO. The results show that all methods benefit from training on MathCog, indicating that  VGPO consistently achieves the best performance among all post-training methods under the same data. This suggests that the performance improvements cannot be attributed to the dataset alone, and that the CogFlow framework and its multi-stage optimization play a central role in the observed gains.
> | **Method** | **Configuration** | **FlowVerse CoT-E (%)** | **FlowVerse Acc (%)** |
> |------------|--------------------|-------------------------|-------------------------|
> | **DAPO** | w/o SVRs, w/o VAR | 56.1 | 45.8 |
> |  | w SVRs, w/o VAR | 59.7 | 50.7 |
> |  | w SVRs, w VAR | 62.8 | 54.1 |
> | **PPO** | w/o SVRs, w/o VAR | 55.9 | 44.6 |
> |  | w SVRs, w/o VAR | 58.2 | 48.3 |
> |  | w SVRs, w VAR | 61.5 | 53.9 |
> | **GRPO** | w/o SVRs, w/o VAR | 56.7 | 47.6 |
> |  | w SVRs, w/o VAR | 61.4 | 52.9 |
> | | w SVRs, w VAR | 64.4 | 55.1 |
> | **VGPO** | w/o SVRs, w/o VAR | 57.4 |48.7 |
> | | w SVRs, w/o VAR | 63.2 | 54.7 |
> | | **w SVRs, w VAR** | **66.0** | **56.2** |
>
> We have expanded the analysis in **Appendix E.4** to better disentangle the  source of performance improvements.
>
>
> ## **Weakness3: Misalignment with cognitive science framing.**
>
> We acknowledge that the mapping between these cognitive-science concepts and the proposed components is not strictly alignment, and we therefore take this opportunity to clarify the intended correspondence.
>
> **First, the “watching” stage in CogFlow is not intended to correspond to internalization**. Its function is limited to extracting low-level structured visual information from an image—such as points, lines, circles, spatial groupings, and relative configurations. These representations are still purely perceptual: they serve as a normalized, noise-reduced description of the raw visual scene, but they do not yet constitute conceptual understanding or task-oriented reasoning.
>
> **The subsequent internalization stage is distinct and is designed precisely to bridge perception and reasoning.** It transforms the structured primitives produced in the watching stage into a conceptually meaningful representation, integrating visual evidence with symbolic relations and problem-specific abstractions. Internalization therefore plays the role of aligning perceptual inputs with the evolving reasoning state.
>
> Under this framing, the reviewer’s observation about error types becomes more interpretable.
>
> - Type 1 (Omit or misbind primitives) errors stem from incorrect or incomplete internalization of otherwise correctly perceived primitives. They reflect failures in mapping the structured perceptual tokens into a coherent conceptual state—hence they lie at the interface between perception and reasoning.
>
> - Type 4 (Invoke external theorems inappropriately) and Type 5 (Refer inconsistently to established elements) arise when the internalized conceptual representation is misinterpreted, misapplied, or inconsistently used within multi-step reasoning. These error types are characteristic of reasoning drift under an inadequately grounded internal state.
>
> We have provided a detailed explanation of this apparent misalignment with the cognitive-science framing in **Appendix E.5**, in order to clarify the conceptual soundness and rationale of our motivation and method.
>
> [4] Roussy M, Mendoza-Halliday D, Martinez-Trujillo J C. Neural substrates of visual perception and working memory: two sides of the same coin or two different coins?[J]. Frontiers in neural circuits, 2021, 15: 764177.
>
> [5] Fuster J M. Cognitive networks (Cognits) process and maintain working memory[J]. Frontiers in neural circuits, 2022, 15: 790691.
>
> [6] Oberauer K. Working memory and attention–A conceptual analysis and review[J]. Journal of cognition, 2019, 2(1): 36.

---

> ### Author Response · Authors · 2025-11-27
> **Rebuttal by Authors [4/8]**
>
> ## **Weakness4: Missing comparison with SophiaVL-R1 and ablations for SVRs components and FG-CLIP.**
> We acknowledge that the original submission Missing comparison with SophiaVL-R1 and ablations for individual SVR components (VPR and VSR), as well as evaluations of how good FG-CLIP is for similarity reward. To address the reviewer’s concerns, we have added the following experiments:
>
> - **Comparison with SophiaVL-R1.** SophiaVL-R1 remains a strong baseline across multimodal reasoning tasks, but on both the MathVerse (CoT-E) and FlowVerse (CoT-E) benchmarks we observe clear performance differences between the two models. **CogFlow-7B surpasses SophiaVL-R1-7B in almost all evaluation configurations, with especially notable gains in text-centric and text–vision balanced settings.** SophiaVL-R1 shows an advantage only in the Vision Only configuration of MathVerse, confirming its strength in purely visual scenarios. **Overall, however, CogFlow-7B achieves consistently higher averages across both benchmarks, indicating a more robust capability in broad visual–mathematical reasoning, particularly when textual understanding and cross-modal integration are required.** These results are now reported in the revised **Tables 1-2**.
>
>   ### MathVerse (CoT-E)
>
>   | Model              | All  | Text Dominant | Text Lite | Vision Intensive | Vision Dominant | Vision Only |
>   |--------------------|------|----------------|-----------|------------------|------------------|-------------|
>   | Multimath-7B       | 26.9 | 28.1           | 15.0      | 25.9             | 34.8             | 30.8        |
>   | DVLR-14B           | 48.1 | 54.3           | 49.0      | 46.3             | 47.2             | 43.8        |
>   | SophiaVL-R1-7B     | 48.8 | 45.4           | 43.9      | 45.1             | 58.5             | **51.3**    |
>   | CogFlow-7B         | **53.9** | **60.7** | **51.2** | **55.0**         | **58.7**         | 44.2        |
>
>
>   ### FlowVerse (CoT-E)
>
>   | Model                     | All  | Text Centric | Text Limited | Text Plus | Vision Dense | Vision Centric | Vision Primary |
>   |---------------------------|------|--------------|--------------|----------|-------------|----------------|----------------|
>   | MultiMath-7B              | 45.2 | 50.6         | 49.9         | -        | 41.7        | 47.2           | 39.7           |
>   | VLM-R1-7B     | 50.7 | 59.0         | 57.9         | 65.5     | 36.2        | 46.1           | 30.6           |
>   | SophiaVL-R1-7B            | 62.3 | 62.7         | 58.9         | 67.7     | 54.8        | 63.5           | 65.6           |
>   | CogFlow-7B                | **66.0** | **67.9** | **67.3** | **68.1** | **57.8** | **68.2** | **66.7** |
>
>
>
>
> - **Ablations on individual SVR components (VPR and VSR).** We note that the Figure 5 in the original submission already included ablations of each SVR component. However, we did not previously investigate how different mixing ratios between VPR and VSR affect performance.
>
>   In the revised manuscript, we therefore conduct an ablation study on the FlowVerse dataset by varying the weighting coefficient $\alpha$ while keeping VAR and the visual gate enabled. The SVR score is defined as
>
>   $$
>   \mathcal{S}\_{\mathrm{SVRs}}=  \alpha\underbrace{\exp\left(-\tfrac{1}{|\mathcal H|}\sum_{(i,j)\in\mathcal H}  \mathcal C(\mathcal{P}\_i,\mathcal{G}\_j)\right)}\_{\mathcal{S}\_{\mathrm{VPR}}} +
>   (1-\alpha)\underbrace{\tfrac{1+\cos\big(\phi(\tilde I),\,\phi(I)\big)}{2}}\_{\mathcal{S}\_{\mathrm{VSR}}}.
>   $$
>
>   Thus, $\alpha = 0$ corresponds to using only the semantic similarity term (VSR), whereas $\alpha = 1$ corresponds to using only the parameter-level term (VPR). **The results show that performance peaks at $\alpha=0.6$, indicating that balancing VPR and VSR yields the most effective supervisory signal.** Moreover, the fact that $\alpha=0$ underperforms $\alpha=1$ suggests that parameter-level supervision (VPR) contributes more substantially to perceptual fidelity than semantic similarity alone (VSR). We provide a more detailed analysis of these ablation results in **Appendix E.9**.
>
>
>
> | α              | 0.0      | 0.2      | 0.4      | 0.6      | 0.8      | 1.0      |
> |----------------|----------|----------|----------|----------|----------|----------|
> | FlowVerse CoT-E / Acc (%)| 63.8/55.3| 64.5/56.0| 65.4/56.9| **66.0/57.2**| 65.4/56.8| 65.2/56.6|

---

> ### Author Response · Authors · 2025-11-27
> **Rebuttal by Authors [5/8]**
>
> - **Evaluating FG-CLIP for similarity reward.**  We further evaluate the FG-CLIP–based similarity module by replacing it with several alternative pre-trained models (e.g. MetaCLIP2[7]) of comparable capacity. **Across these variants, the FG-CLIP (ViT-L-14) configuration consistently achieves the best overall performance on FlowVerse, indicating that FG-CLIP provides a stronger and more discriminative supervision signal for our VSR component than the alternative encoders.** We provide a more detailed analysis of these ablation results in **Appendix E.10**.
>
> | Pre-trained Model      | MetaCLIP2(ViT-B-16)  | MetaCLIP2(ViT-L-14)  | FG-CLIP(ViT-B-16) | FG-CLIP(ViT-L-14) |
> |-----------------|------|------|------|------|
> | FlowVerse CoT-E / Acc (%)     | 62.5/53.9|63.2/54.5|63.3/54.6|**63.8/55.3**|
>
> [7] Chuang Y S, Li Y, Wang D, et al. Meta clip 2: A worldwide scaling recipe[J]. arXiv preprint arXiv:2507.22062, 2025.
>
> ## **Additional suggestions**
>
> ### 1. **Spelling and Formatting Typos.**
> We apologize that the original submission contained these typographical and labeling inaccuracies.
>
> All issues pointed out by the reviewer—including the misspellings (“grated”→“gated”, “that produce appear”→“that appear”) and the mislabeling in Figure 3 (“Visual Similarity Reward” → “Visual Semantic Reward”, and “Visual Parameter Reward” → “parameterized”).
>
> We have now been fully corrected in the revised manuscript.
>
> ### 2. **Lots of acronyms and some acronyms aren't very descriptive.**
>
> We apologize that the extensive use of acronyms in the original submission (e.g., SVRs, VAR, VGPO, VPR, VSR) may have caused unnecessary confusion.
>
> We will streamline the use of acronyms and revise several of them to improve readability. To avoid potential confusion for other reviewers during the rebuttal phase, we will finalize these acronym changes in the camera-ready version. For example, we plan to replace SVRs (Synergistic Visual Rewards) with the more descriptive SynVRs.
>
>
> ### 3. **Equation 4 does not follow the algorithm as described.**
> We apologize that Equation (4) in the original submission did not faithfully reflect the VGPO algorithm. We have rewritten the equation to improve clarity and to ensure full consistency with the algorithmic procedure used in our implementation.
>
> Specifically, given a perception candidate $y _i^{\mathrm{w}}(x)$
> and the ground-truth $\hat{y}^{\mathrm{w}}(x)$, while $\tilde I_i$ and $I$ denote their respective renderings, we define the perceptual accuracy score as:
>
> $$
> S\_{\mathrm{vis}}\big(y\_i^{\mathrm{w}}(x)\big)=
> \begin{cases}
>  \mathcal{S}\_{\mathrm{VPR}}\big(y\_i^{\mathrm{w}}(x),\,\hat{y}^{\mathrm{w}}(x)\big) +  \mathcal{S}\_{\mathrm{VSR}}\big(\tilde I\_i,\, I\big), & \text{training,}
> \\\\
>  \mathcal{S}\_{\mathrm{VSR}}\big(\tilde I\_i,\, I\big), & \text{inference.}
> \end{cases}
> $$
>
>
> Subsequently, the visual gate $\Gamma(\cdot)$ enforces perception quality by scoring each perception trajectory and accepting the first attempt whose score exceeds a preset threshold. If no attempt passes within $M$ trials, the gate returns the attempt with the highest $S\_{\mathrm{vis}}$:
>
> $$
> \kappa =
> \begin{cases}
> \min\{\,k\in\{1,\dots,M\}:\ S_{\mathrm{vis}}(y_{i,k}^{\mathrm{w}}(x)) \ge \tau\,\}, & \{k:\, S_{\mathrm{vis}}(y_{i,k}^{\mathrm{w}}(x)) \ge \tau\} \neq \varnothing,\\\\
> M, &  \{k:\, S_{\mathrm{vis}}(y_{i,k}^{\mathrm{w}}(x)) \ge \tau\} = \varnothing.
> \end{cases}
> $$
>
> $$
> y\_i^{\mathrm{w}*}(x) = \Gamma\big(\{y\_{i,k}^{\mathrm{w}}(x)\}\_{k=1}^{\kappa}\big)
> = \operatorname\*{arg\,max}\_{1\le k\le \kappa} S\_{\mathrm{vis}}\\big(y\_{i,k}^{\mathrm{w}}(x)\big).
> $$
>
>
> where $\kappa$ is the stopping index, defined as the smallest $k$ whose score reaches acceptance threshold $\tau$, $y_i^{\mathrm{w}*}(x)$ is the perception trajectory selected by the visual gate, and $y_{i,k}^{\mathrm{w}}(x)$ is the $k$-th perception trajectory for input $x$.
>
> We have rewritten Equation (4) to improve clarity and to ensure full consistency with the VGPO algorithm described in the manuscript.
>
> ### 4. **Equation 3 should include preferred trajectory in the denominator sum.**
> We apologize that Equation (3) in the original submission did not include the preferred trajectory in the denominator sum. We have rewritten the equation to improve clarity and to ensure full consistency with the mathematical formulation of Softmax-DPO[7]:
>
>   $$
>   \mathcal{L}_{\text{S-DPO}}= - \log \sigma\left(- \log \sum\_{j=1}\^{m} \exp\big( s\_j\^{-} - s\^{+} \big)\right), \quad   s = \beta \big[ \log \pi\_\theta(y \mid x) - \log \pi\_{\text{ref}}(y \mid x) \big],$$
>
> where $s^{+}$ denotes the score of the preferred trajectory, $\{s_j^{-}\}_{j=1}^m$ represent the corresponding scores of the dispreferred trajectories, and $\sigma(\cdot)$ denotes the sigmoid function.
>
> We have rewritten Equation (3) to improve clarity and update the citation to Softmax-DPO[8] in revised manuscript **Section 3.2**.

---

> ### Author Response · Authors · 2025-11-27
> **Rebuttal by Authors [6/8]**
>
> ### **5. Missing the details of SFT stage is unclear in Figure 2.**
> We acknowledge that the original submission did not clearly describe the SFT stage in Figure 2, and that the yellow/green outlines may have unintentionally suggested that Watching and Thinking are only used during RL. We clarify the SFT stage as follows.
>
> Both the MathCog-SFT and MathCog-RL splits are organized into three components—“Watching,” “Thinking,” and “Answer”.  **In our training pipeline, we first perform supervised fine-tuning (SFT) on the MathCog-SFT,** where the model is trained on the “Watching” sequences together with the corresponding “Thinking,” and “Answer”  to strengthen its basic perceptual and reasoning abilities.
>
> We have now provided a detailed description of the SFT stage in **Appendix D.5** and **Figure 10**, and we have redrawn **Figure 2** to avoid potential misinterpretation.
>
> ### **6. One-sentence Captions need to be expanded.**
>
> We apologize that the original submission lacked detailed captions for the figures and tables. We have expanded all one-sentence captions to provide clearer context on the experimental setup and key findings. We have provided detailed descriptions of the figures and tables in the revised manuscript **Section 3.1** and **Section 4.1**.
>
> - **Pipeline (Figure 2)**
> Inspired by the cognitive paradigm, CogFlow introduces a visual-gated policy optimization (VGPO) strategy within the visual gate to explicitly anchor the reasoning process in perceptual accuracy while problem-solving. In conjunction with the Synergistic Visual Rewards (SVRs), Visual-Anchored Reward (VAR) and Inference Reward (IR), VGPO holistically enhances cognitive capability in synchrony.
>
> - **SVRs illustration (Figure 3)**
>   The caption has been expanded to explain that Synergistic Visual Rewards combine VPR and VSR to enforce both local geometric fidelity and global perceptual coherence, and that these two signals together provide a unified supervision mechanism for robust perception.
>
>
> - **VGPO illustration (Figure 4)**
>   The caption now clarifies that VGPO introduces a visual gate and multiple rewards to strengthen multi-step visual reasoning, and explicitly states that it couples perceptual quality control with outcome-based optimization to obtain more stable and interpretable trajectories.
>
>
> [8] Chen Y, Tan J, Zhang A, et al. On softmax direct preference optimization for recommendation[J]. Advances in Neural Information Processing Systems, 2024, 37: 27463-27489.

---

> ### Author Response · Authors · 2025-11-27
> **Rebuttal by Authors [7/8]**
>
> ## **Question1: Why does the Baseline+SVRs setting show limited reduction in perception errors, but more reduction in reasoning errors?**
>
> We acknowledge that the Baseline+SVRs setting shows limited reduction in perception errors, but more reduction in reasoning errors (Figure 7 in the original submission). We would like to explain this phenomenon as follows.
>
>
> **First, we view this as a natural consequence of the fact that perception, internalization, and reasoning are not independent modules**: improvements in perception inevitably propagate to, and influence, the subsequent reasoning process.
> **We conduct a systematic case study to analyze the error compensation mechanism. In the illustrated example, the perception stage exhibits partial errors:** the coordinates of point $A$ are misidentified, which in turn corrupts lines $AB$ and $AD$. Nevertheless, during the subsequent reasoning stage, the model still produces the correct answer by accurately perceiving other visual primitives (e.g., points $B$ and $C$) and performing precise *knowledge internalization*. This analysis reveals a cognitive processing pattern in which response generation proceeds sequentially through **Perception** ⇒ **Internalization** ⇒ **Reasoning**, and suggests that strengthened perception and internalization capabilities can mitigate reasoning errors even in the presence of localized perception failures.
>
> In addition, among the Baseline+SVRs, Baseline+VAR, and Baseline+VGPO configurations, **Baseline+SVRs already yields the largest reduction in perception errors, further demonstrating that SVRs provide effective improvements to the perceptual component.**
>
> We have added this analysis to **Appendix E.11 and Figure 18**, in order to provide a more detailed explanation of the error compensation mechanism.

---

> > ### Author Response · Authors · 2025-12-01
> > **Rebuttal by Authors [8/8]**
> >
> > ## **Question2: What is the effect of the Visual Gate and VGPO at Inference?**
> >
> > We thank the reviewer for this insightful question, which prompted us to conduct a more fine-grained analysis of the role of the visual gate and VGPO at inference time. We clarify this below by explicitly disentangling the effect of the visual gate at inference from the effect of VGPO.
> >
> >
> >
> >   To ensure a fair comparison, all results reported in Table 4 in the original submission are obtained with **the visual gate enabled during inference**. That is, for every variant we use the same inference pipeline — perception sampling with the gate, followed by a single reasoning pass — and only the training configuration (SVRs / VAR / VGPO) is changed.
> >
> >
> >
> >   **To disentangle the effect of VGPO from the effect of the gate itself, we conducted an additional ablation in which the model is trained without** VGPO, but the visual gate is still applied at inference. The results are summarized in the table below.
> >
> >   Concretely, *We conduct an additional ablation study on FlowVerse by varying the training configuration (SVRs / VAR / VGPO / VG)*, where VG represents the visual gate at inference. The results are summarized in the table below.
> >   These results indicate that **(1) visual gating is also beneficial during inference, yielding a consistent absolute gain of around 0.6-1% accuracy even when the model is trained without VGPO**, and **(2) VGPO provides an additional and more substantial improvement, because it uses the visual gate during training to shape the policy itself**, rather than only filtering outputs at test time.
> >
> >   We have provided this analysis and presented the results in the **Table 5**  and **Figure 8** in **Section 4.3**.
> >
> >   | SVRs | VAR | VGPO | VG (Inference) | FlowVerse-CoT | FlowVerse-Acc | MathVerse-CoT | MathVerse-Acc |
> >   |:----:|:----:|:----:|:--------------:|:-------------:|:-------------:|:-------------:|:-------------:|
> >   | ✗ | ✗ | ✗ | ✗ | 56.6 | 47.9 | 47.0 | 34.9 |
> >   | ✗ | ✗ | ✗ | ✔ | 57.4 | 48.7 | 48.2 | 35.6 |
> >   | ✔ | ✗ | ✗ | ✔ | 63.2 | 54.7 | 50.5 | 36.9 |
> >   | ✗ | ✔ | ✗ | ✔ | 62.7 | 53.5 | 49.9 | 36.2 |
> >   | ✗ | ✗ | ✔ | ✔ | 63.4 | 54.8 | 50.8 | 37.3 |
> >   | ✔ | ✔ | ✗ | ✔ | 64.4 | 55.1 | 52.1 | 38.0 |
> >   | **✔** | **✔** | **✔** | **✔** | **66.0** | **56.2** | **53.9** | **39.5** |
> >
> >
> > ## **Question3: What do the easy/medium/hard categories refer to?**
> > We acknowledge that the original submission did not clearly describe the easy/medium/hard categories. We clarify this below:
> >
> > The easy/medium/hard labels characterize the relative difficulty for the model to detect and correct different types of reasoning drift errors, rather than the difficulty of constructing these cases. **Easy errors are typically local and isolated** (e.g., a minor mistake in a single visual primitive or reasoning step) **and can often be corrected using limited contextual information**. **Medium errors involve multiple interacting primitives or steps and require the model to integrate information across a broader context**. **Hard errors are globally entangled with the full reasoning chain, where correcting them demands coherent understanding of both the visual configuration and the multi-step logical structure of the solution.**
> >
> > We have provided a more detailed explanation of the easy/medium/hard categories in **Appendix C.2** and **Figure 11**.

---

### Official Review · Reviewer_joaQ · 2025-10-31

**Soundness:** 3
**Presentation:** 3
**Contribution:** 2
**Rating:** 6
**Confidence:** 3

**Summary:**

The paper presents CogFlow, a cognitively inspired framework for visual mathematical reasoning that models the process through three stages: perception, internalization, and reasoning. It introduces stage-specific visual rewards to strengthen the coupling between visual understanding and symbolic reasoning. A new dataset, MATHCOG, is constructed to support training and evaluation. Experiments on FlowVerse, MathVerse, and MathVista show consistent improvements, demonstrating the effectiveness of multi-stage perceptual guidance.

**Strengths:**

1. The paper presents a conceptually clear and cognitively inspired framework that divides multimodal reasoning into perception, internalization, and reasoning stages, effectively bridging visual understanding and symbolic inference.

2. The paper demonstrates a well-organized methodological design, where the layered reward mechanisms are technically coherent and systematically integrated across different reasoning stages.

3. The paper show consistent improvements on three visual mathematical benchmarks, and contributes a new dataset, MATHCOG, that holds potential value for future research.

**Weaknesses:**

1. The paper mainly reports results on MathVista, where the gains are substantial, but lacks systematic comparisons with other recent benchmarks such as MathVision, We-Math, and DynaMath. This limits the generality of the claimed performance advantage.

2. The paper evaluates only a single model scale (Qwen2.5-VL-7B) without testing different sizes or architectures, making it difficult to assess the scalability and broader applicability of the proposed framework.

3. The paper provides limited analysis of why the multi-stage reward structure improves reasoning performance. The overall explanation remains empirical, and deeper insights into the mechanism behind these gains would strengthen the work's interpretability.

**Questions:**

The author should include additional experimental results, particularly on other recent benchmarks such as MathVision, We-Math, and DynaMath. Since the performance improvement on MathVista is exceptionally large, results on these datasets are important to verify the generality and robustness of the proposed method.

---

> ### Author Response · Authors · 2025-11-27
> **Rebuttal by Authors [1/2]**
>
> We sincerely thank Reviewer `joaQ` for recognizing our **conceptually clear and cognitively inspired framework**, **a well-organized methodological design** and **potential dataset**. In the following, we address the remaining concerns point by point.
>
> ## **Weakness1: The lack of systematic comparisons with other recent benchmarks.**
> We would like to clarify that the evaluation in our initial submission (Tables 1–3) was already conducted on three challenging benchmarks that span a wide range of problem types, diagram styles, and difficulty levels. In the revised version, to provide a more comprehensive and convincing demonstration of the generality and robustness of CogFlow’s performance, **we further strengthen the empirical study by adding three additional benchmarks: WeMath, LogicVista, and DynaMath.**
>
> In particular, among open-source models, CogFlow-7B achieves the best performance on WeMath and DynaMath, and ranks second on LogicVista, slightly behind GLM-4.1V-9B. Compared to InternVL3-8B, CogFlow yields large absolute gains. Moreover, CogFlow substantially narrows the gap to strong closed-source systems such as GPT-5 and Gemini-2.5-Pro, **indicating that the proposed framework brings robust benefits across diverse visual–mathematical benchmarks.**
>
> | Models            | WeMath | LogicVista | Dynamath |
> |-------------------|--------|-----------|----------|
> ||Close-Source|||
> | GPT-5             | 71.1   | 70.0      | *60.9*     |
> | Claude-3.7-Sonnet | 49.3   | 58.2      | 39.7     |
> | Gemini-2.5-Pro    | *78.0*   | *73.8*      | 56.3     |
> ||Open-Source|||
> | Keye-VL-8B        | 60.7   | 54.8      | 37.3     |
> | InternVL3-8B      | 37.1   | 44.1      | 25.5     |
> | GLM-4.1V-9B       | 63.8   | **60.4**      | 42.5     |
> | CogFlow-7B        | **64.1**   | 58.1      | **46.2**     |
>
> We have also provided the detailed evaluations on multiple benchmarks in **Table 4**, which further confirm the generality of CogFlow's performance advantage.
>
>
> ## **Weakness2: Lack of testing different sizes or architectures.**
> We recognize that the previous version did not include evaluations across different model sizes or architectures, which limited the assessment of CogFlow’s scalability and architectural generality.
>
> To demonstrate the scalability and broader applicability of CogFlow, we now explicitly evaluate CogFlow across **multiple backbone architectures and parameter scales**, rather than only on Qwen2.5-VL-7B. Specifically, we report results for **Qwen2.5-VL (3B and 7B) and InternVL3.5 (2B, 4B, and 8B)**. The results show that: (1) CogFlow consistently outperforms models of comparable size, and even with only 2B or 4B parameters it achieves competitive performance. For example, CogFlow surpasses strong baselines such as SophiaVL-R1-7B, which achieves 47.9 / 38.7 on FlowVerse, indicating that CogFlow attains better performance on both MathVerse and FlowVerse.
> (2) the performance gains are maintained or even amplified as model capacity increases;
> (3) the improvements hold for both the Qwen and InternVL families, indicating that the framework is not tied to a specific architecture.
> **These findings provide empirical evidence that CogFlow is scalable and broadly applicable beyond a single model configuration.**
>
>
>   | **Models** | **Qwen2.5-VL-3B** | **Qwen2.5-VL-7B** | **InternVL3.5-2B** | **InternVL3.5-4B** | **InternVL3.5-8B**|
> |-----------|--------------------|--------------------|---------------------|---------------------|---------------------|
> | **FlowVerse** | 60.4 / 51.8 | 66.0 / 56.2 | 58.2 / 47.1 | 62.5 / 54.7 | 66.3 / 56.9 |
> | **MathVerse** | 48.8 / 37.6 | 53.9 / 39.5 | 47.2 / 35.8 | 52.2 / 38.3 | 54.2 / 40.0 |
>
> We summarize these results across different architectures and model sizes in **Table 5**, which further supports the scalability and broad applicability of CogFlow.

---

> ### Author Response · Authors · 2025-11-27
> **Rebuttal by Authors [1/2]**
>
> ## **Weakness3: Limited insights into the mechanism of multi-stage reward structure.**
> We acknowledge that our previous analysis of how the multi-stage reward structure improves CogFlow’s reasoning performance (Section 4.3, Appendix E, and Figures 5–8 and 13 in the initial submission) was not sufficiently clear. In the revised manuscript, we provide a more detailed and systematic discussion of the mechanisms through which the multi-stage reward structure enhances reasoning performance.
>
> **On the one hand, each reward is designed to target a distinct failure mode in visual mathematical reasoning**:
> 1. **The Synergistic Visual Rewards (SVRs), which comprise VPR and VSR, are designed to address perceptual failures.** In the revised manuscript, Figure 7 presents an ablation of the SVR components. The results show that adding either VPR or VSR alone consistently improves performance, and the best results are obtained when both are enabled. Figure 13 further reports the perception F1 scores for different SVR variants, indicating that removing either reward degrades perception fidelity. Finally, the error-type analysis in Figure 10 demonstrates that SVRs effectively reduce perception-related errors, thereby validating their effectiveness. Taken together, these results show that VPR and VSR provide complementary supervision signals: VPR enforces local geometric fidelity, while VSR encourages global perceptual coherence in terms of overall style and layout, and together they form trustworthy visual cues that serve as a robust foundation for effective visual mathematical reasoning, which is consistent with the conclusions reported in prior studies[1, 2].
>   2. **The Visual-Anchored Reward (VAR) is designed to alleviate reasoning drift by bridging the perception and reasoning stages, encouraging the model to produce structured, reasoning-ready outputs** (i.e., knowledge-internalized representations [3]) that provide a more reliable foundation for subsequent reasoning. In the revised manuscript, the ablation study on VAR (Figure 14) shows that removing any single error type consistently degrades performance, indicating that each error type provides complementary supervision. The largest drops occur when excluding the omission/misbinding primitives or contradicting geometric constraints error types, highlighting that correctly binding primitives and respecting core geometric constraints are most critical for keeping reasoning tied to perception. In addition, the error-type analysis in Figure 10 further demonstrates that VAR effectively reduces knowledge internalization errors, which in turn leads to improved overall performance.
>
>   3. **The inference reward (IR) ensures task-level correctness and proper output structure** at the final reasoning stage[4].
>
> **On the other hand, this multi-stage reward structure provides a principled solution to the credit-assignment problem**: instead of relying on a single sparse task-level reward, the stage-wise signals supply informative gradients at different points of the perception–internalization–reasoning pipeline.
> The ablation on CogFlow’s components (Table 5 in the revised manuscript) shows that both SVRs and VAR are essential for achieving strong performance, and that the best results are obtained when all rewards are enabled. In addition, the reasoning-drift analysis across three representative pipelines (Figure 1(b) in the revised manuscript) demonstrates that CogFlow’s reasoning trajectories are more stable and better aligned with the visual information, which is consistent with the observed performance gains.
> Moreover, the error-type analysis in Figure 9 shows that, compared with the Baseline+VGPO setting, CogFlow substantially reduces perception errors, knowledge-internalization errors, and reasoning errors. **This indicates that the three rewards together form a coherent cognitive paradigm: improved perception supports more reliable internalization, and VAR further aligns the reasoning trajectory with the visual evidence**
>
> We have provided a more detailed analysis of the multi-stage reward structure in the revised manuscript (**Section 4.3, Appendix E.6, and Figures 8–9 and 14–15**), which offers deeper insights into the mechanisms behind the observed gains and strengthens the interpretability of our approach.
>
> [1] Guo Z, Liu M, Wang Q, et al. Integrating Visual Interpretation and Linguistic Reasoning for Math Problem Solving[C]. ICCV, 2025.
>
> [2] Chen F, Yuan H, Xu Y, et al. MathFlow: Enhancing the Perceptual Flow of MLLMs for Visual Mathematical Problems[J]. arXiv preprint arXiv:2503.16549, 2025.
>
> [3] Ryan R M, Connell J P. Perceived locus of causality and internalization: examining reasons for acting in two domains[J]. Journal of personality and social psychology, 1989, 57(5): 749.
>
> [4] Shao Z, Wang P, Zhu Q, et al. Deepseekmath: Pushing the limits of mathematical reasoning in open language models[J]. arXiv preprint arXiv:2402.03300, 2024.

---

### Official Review · Reviewer_xQ8R · 2025-11-01

**Soundness:** 4
**Presentation:** 2
**Contribution:** 3
**Rating:** 6
**Confidence:** 4

**Summary:**

The paper argues that the existing vision-language reasoners lack high-quality perception and their reasoning drifts away from their perception in their chain-of-thought. To address these issues, the paper proposes a Cogflow framework that encourages the model to internalize the perceptual primitives before it starts solving the problem. Specifically, the paper proposes visual rewards to ensure that perceptual accuracy in parameter space and layout consistency. This acts as a gate to prevent low-quality generations from unrolling and training rewards for RL training. Further, the paper proposes visual anchored rewards which encourages the model to internalize the perception in its chain of thoughts through synthetically generated negative pairs for softmax-DPO. Finally, they propose visual-gated policy optimization (VGPO) as a global optimization of the model policy. As a result, the model starts performing very well on the FlowVerse and MathVerse datasets. The paper also studies the impact of different design choices in ablation studies.

**Strengths:**

1. The paper addresses an important problem of the lack of perceptual understanding and reasoning drift in VLMs.

2. The proposed method is quite comprehensive where each component ensures that the model learns to perceive images accurately at the primitive level and the model is encouraged to utilize perception via additional rewards.

3. The experimental results suggest that the method works very well on the FlowVerse, MathVerse, and MathVista dataset. The ablation studies showcase the usefulness of different components on final performance.

**Weaknesses:**

1. Since the paper has too many moving parts, the paper could be written better. Despite decent experience in VL reasoning, I was finding it hard to keep up with a dense introduction with many jargons. The authors should think about how to portray their story in a more simplified manner.

2. The proposed method to get the primitives for shapes (e.g., Circle) does not seem scalable. How will you get primitives for shapes where the actual dimensions are absent like many natural scenes or even synthetic scenes (like CLEVR dataset)? There is very little information about collecting “watching” data in the main text. Overall, the solution is more tailored towards geometry than general-purpose visual reasoning. It is critical to acknowledge that early in the paper to set the expectations right.

3. The evaluation is somewhat limited. It would be nicer to show performance on more evaluation datasets such as LogicVista, MathVision, We-Math, MMMU-Pro, HallusionBench. Currently, my hunch is that the method will suffer on very complex scenes from LogicVista because there are many images within an image and getting those many primitives might break the model.

**Questions:**

mentioned above

---

> ### Author Response · Authors · 2025-11-27
> **Rebuttal by Authors [1/2]**
>
> We sincerely thank Reviewer `xQ8R` for recognizing our **contribution to addressing reasoning drift in VLMs**, **comprehensive method for ensuring accurate perception** and **useful ablation studies**. In the following, we address the remaining concerns point by point.
>
> ## **Weakness 1: Narrative is overly dense with too many moving parts and jargon.**
> We apologize that the previous manuscript contained many moving parts and that the dense introduction with extensive jargon (in the introduction and methodology sections of the initial submission) may have made it difficult for the readers to fully follow the narrative.
> In response, we have made several key revisions to improve the clarity and readability of the paper:
>
> ### 1.1 **Clarification and simplification of the narrative**
> **We provide a clearer context for why the problem is significant and how it motivates the development of our approach, COGFLOW.** In particular, we explicitly highlight the main challenge faced by existing VLMs in solving visual mathematical problems—namely, the *reasoning drift* issue, where models fail to maintain alignment between visual perception and symbolic reasoning, resulting in low answer accuracy and inconsistent reasoning chains. Building on this clarified problem statement, we also streamline the exposition of CogFlow’s three-stage pipeline (perception, internalization, and reasoning), so that readers can first grasp the core conceptual flow before engaging with the technical details. This restructuring helps ensure that the key ideas are clear and accessible to a broader audience, while preserving the full technical depth of the framework.
>
> ### 1.2 **Streamlining of technical jargon**
> **We have actively reduced the use of overly technical jargon** when introducing our methodology and added more intuitive, reader-friendly explanations to improve accessibility. In particular, **we aimed to make the core concepts of our approach more accessible by simplifying complex terminology**. For example,  we now describe the Visual Parameterized Reward (VPR) as a measure of geometric accuracy and the Visual Semantic Reward (VSR) as a measure of overall style and layout consistency when they are first introduced.
>
>
> We have largely revised the manuscript (especially **Sections 1 & 3**) to make it clearer and more accessible by addressing the complexity and jargon that may have hindered understanding.

---

> ### Author Response · Authors · 2025-11-27
> **Rebuttal by Authors [2/2]**
>
> ##  **Weakness2: The proposed method to get the primitives does not seem scalable.**
> We apologize for the insufficient and somewhat vague clarification of how primitives are obtained in our initial submission. In the revised version, we provide a more detailed and explicit explanation of the scalability of our primitive.
>
> While our work is primarily focused on visual mathematical problems, the procedure for obtaining these primitives is domain-agnostic. **The notion of primitives, as a structured representation of visual information, naturally extends to natural scenes beyond the domain of shapes.**
> For example, in natural scenes, a single image may contain objects such as people, trees, buildings, and sky regions, along with their visual attributes like colors and spatial relations.
>
> We first use an object detection model or a segmentation model to identify instances such as 'person <x1, y1, x2, y2> in front of car <x3, y3, x4, y4>' or 'tree <x5, y5, x6, y6> next to building <x7, y7, x8, y8>'. With the assistance of an LLM and human annotators, these elements can then be converted into structured visual information (visual primitives), for example: 'Person: coordinates: <x1, y1, x2, y2>, sex: male, clothing: red shirt; Tree: coordinates: <x5, y5, x6, y6>, type: oak, height: 5 meters; Car: coordinates: <x3, y3, x4, y4>, make: Toyota, model: Corolla, color: blue; Building: coordinates: <x7, y7, x8, y8>, type: residential, floors: 3'.
>
> We have provided a more detailed explanation of the scalability of obtaining the primitives in **Appendix B**, which further clarifies their potential applicability to scenes that extend beyond simple geometric shape.
>
> ## **Weakness3: The evaluation is somewhat limited.  It would be nicer to show performance on more evaluation datasets.**
> We would like to clarify that the evaluation in our initial submission (Tables 1–3) was already conducted on three challenging benchmarks that span a wide range of problem types, diagram styles, and difficulty levels. In the revised version, to provide a more comprehensive and convincing demonstration of the generality and robustness of CogFlow’s performance, we further evaluate it on three additional benchmarks: WeMath[1], LogicVista[2], and DynaMath[3].
>
> We have conducted additional evaluations on **WeMath, LogicVista, and Dynamath**, comparing CogFlow with a range of open- and closed-source baselines. As shown in the table below, CogFlow consistently outperforms models of comparable scale, achieving the best performance on WeMath and Dynamath, and competitive results on LogicVista. We note that MathCog primarily consists of structured data, which introduces a gap when reasoning over benchmarks with more irregular or unstructured images. **Nonetheless, it is reasonable to expect that incorporating more unstructured training data would further improve CogFlow’s performance on these benchmarks.**
>
> | Models            | WeMath | LogicVista | Dynamath |
> |-------------------|--------|-----------|----------|
> ||Close-Source|||
> | GPT-5             | 71.1   | 70.0      | *60.9*     |
> | Claude-3.7-Sonnet | 49.3   | 58.2      | 39.7     |
> | Gemini-2.5-Pro    | *78.0*   | *73.8*      | 56.3     |
> ||Open-Source|||
> | Keye-VL-8B        | 60.7   | 54.8      | 37.3     |
> | InternVL3-8B      | 37.1   | 44.1      | 25.5     |
> | GLM-4.1V-9B       | 63.8   | **60.4**      | 42.5     |
> | CogFlow-7B        | **64.1**   | 58.1      | **46.2**     |
>
> We have also provided detailed evaluations on multiple benchmarks in **Tables 3** in revised manuscript, which further confirm the superiority of CogFlow.
>
> [1] Qiao R, Tan Q, Dong G, et al. We-math: Does your large multimodal model achieve human-like mathematical reasoning?[C]//Proceedings of the 63rd Annual Meeting of the Association for Computational Linguistics (Volume 1: Long Papers). 2025: 20023-20070.
>
> [2] Xiao Y, Sun E, Liu T, et al. Logicvista: Multimodal llm logical reasoning benchmark in visual contexts[J]. arXiv preprint arXiv:2407.04973, 2024.
>
> [3] Zou C, Guo X, Yang R, et al. Dynamath: A dynamic visual benchmark for evaluating mathematical reasoning robustness of vision language models[J]. arXiv preprint arXiv:2411.00836, 2024.

---

### Official Review · Reviewer_gjDa · 2025-11-04

**Soundness:** 2
**Presentation:** 3
**Contribution:** 3
**Rating:** 4
**Confidence:** 4

**Summary:**

This paper proposes a novel cognitive-inspired three-stage framework (perception-internalization-reasoning) to enable faithful integration and proper utilization of visual signals for visual mathmatical reasoning.  The authors proposes synergistic vision reward (SVR) consisting of visual parameterized reward and the visual semantic reward and incoporate the reward in the original GRPO algorithm to devise the VRPO algorithm. The author proposes a new visual anchor reward and curates constrastive training examples to mitigate five common types of unfaithful ultilization of visual signals.

**Strengths:**

+ The paper proposes a new three-step framework for faithful visual mathematical reasoning.
+ It constructs a new dataset MATHCOG featured with three subset with each subset for a stage respectively.
+ Experiments on FlowVerse dataset and MathVerse dataset show substaintial improvement compared with baseline methods.

**Weaknesses:**

+ Some important details are missing. For example, how to construct the contrastive pairs in MATHCOG-VAR dataset? How to incorporate the five common types of reasoning error? How is the visual anchor reward $R_{VAR}$ implemented? Is it a reward in VRPO algorithm or is it a training stage between SFT and VRPO? How iis the correctness and format reward $R_{IR}$ implemented? How to compute the visual parameterized reward in the parameter space if the predicted primitive and the ground-truth primitive is not within the same class? (i.e., the ground truth is an ellipse while the predicted primitive is a circle).

+ As in Eq4, the generation of perception output is repeated $k$ times until the SVR reward is large than a fixed threshold. It is eqivalent to best-of-k sampling if the threshold is not met in early attempts. Therefore, it is doubtful whether the comparison with baseline method is fair or not.

**Questions:**

See the weaknesses above.

---

> ### Author Response · Authors · 2025-11-27
> **Rebuttal by Authors [1/4]**
>
> We sincerely thank Reviewer `gjDa` for recognizing our **novel visual mathematical reasoning framework**, the **dataset contributions** and **substantial performance improvement**.
> Below, we address the remaining concerns point by point.
>
> ## **Weakness1: Some important details are missing.**
>
> ### **1.1 How to construct the contrastive pairs in MathCog-VAR dataset?**
>
> We apologize that the previous description of the MathCog-VAR construction (Appendix C.2.2 and Figures 9 & 10 in our initial submission) lacked sufficient clarity. We have rewritten and reorganized this part to provide a clearer and more detailed account of the construction process, particularly the generation of contrastive pairs.
>
> Specifically, the construction pipeline can be summarized in the following steps:
> 1. We first sample 10,000 examples as positive samples from  MathCog-SFT.
> 2. We categorize the errors into five types: (1) omitting or misbinding primitives, (2) introducing nonexistent facts, (3) contradicting geometric constraints, (4) invoking external theorems inappropriately, and (5) referring inconsistently to established elements. We use an LLM (i.e., GPT-5) to generate negative trajectories conditioned on each positive trajectory by explicitly instructing it to modify the corresponding *Thinking* part, as specified in the prompt shown in the table below. Further details can be found in **Table 8** of the revised paper.
>
>
> | **Error Type**                                | **Instruction**                                                                                                                                                                                                                                                                             | **Positive Data** |
> |-----------------------------------------------|---------------------------------------------------------------------------------------------------------------------------------------------------------------------------------------------------------------------------------------------------------------------------------------------|-------------------|
> | Omit or Misbind Primitives                    | *Starting from the correct reasoning, deliberately omit or misbind at least one basic geometric primitive (points, lines, circles, etc.), while keeping the overall reasoning fluent and locally plausible.*                                                                               | {positive data}   |
> | Introduce Nonexistent Facts                   | *Starting from the correct reasoning, introduce at least one geometric or numerical fact that is not supported by the given figure or problem statement, but make the reasoning appear locally coherent and natural.*                                                                       | {positive data}   |
> | Contradict Geometric Constraints              | *Starting from the correct reasoning, modify the chain of thought so that at least one step violates the true geometric constraints (e.g., equal lengths, parallelism, angle measures), while preserving a seemingly reasonable narrative.*                                                 | {positive data}   |
> | Invoke External Theorems Inappropriately      | *Starting from the correct reasoning, inappropriately invoke at least one external theorem or formula whose preconditions are not satisfied, or whose use is not justified by the internalized structure, while keeping the explanation linguistically smooth.*                            | {positive data}   |
> | Refer Inconsistently to Established Elements  | *Starting from the correct reasoning, alter the chain of thought so that references to previously established elements (points, lines, or relationships) become inconsistent, such as swapping labels or changing properties across steps, but without breaking the overall fluency of the text.* | {positive data}   |
>
>
>
> 3. We conduct a round of human verification, during which annotators manually review and correct samples that do not meet the required criteria. This process results in 10,000 positive samples and 50,000 negative samples. Throughout the curation period, we employ a team of **31 annotators** who work over the course of **16 days** to refine and validate all generated candidates.
>
> **Overall, the principle of curation is to generate negative trajectories by modifying the *Thinking* part of the original positive trajectory, and then manually check and filter the generated candidates.** We have expanded our description of the curation process in **Figures 10-11** and **Appendix C.2** of the revised paper, which clearly illustrate how the contrastive pairs and the five error types are constructed.

---

> ### Author Response · Authors · 2025-11-27
> **Rebuttal by Authors [2/4]**
>
> ### **1.2 How to incorporate the five common types of reasoning error?**
> After obtaining the MathCog-VAR dataset, we effectively incorporate the five common types of reasoning errors by adopting Softmax-DPO. This method provides a more stable and comprehensive learning signal compared to pairwise DPO by jointly normalizing over multiple dispreferred samples and reducing reward sparsity. Formally, the objective is:
>
>   $$
>   \mathcal{L}_{\text{S-DPO}}= - \log \sigma\left(- \log \sum\_{j=1}\^{m} \exp\big( s\_j\^{-} - s\^{+} \big)\right), \quad   s = \beta \big[ \log \pi\_\theta(y \mid x) - \log \pi\_{\text{ref}}(y \mid x) \big],$$
>
> where $s^{+}$ denotes the score of the preferred trajectory, $\{s_j^{-}\}_{j=1}^m$ represent the corresponding scores of the dispreferred trajectories, and $\sigma(\cdot)$ denotes the sigmoid function.
>
> **After training, the VAR model is equipped with the ability to detect 5 reasoning errors in a trajectory and assigns a scalar reward between 0 and 1.** We have expanded our description of the five common types of reasoning error incorporation in **Appendix C.2.2** and **Section 3.2** in the revised paper, which clearly illustrate how the VAR model is implemented.
>
>
> ### **1.3 How is the visual-anchored reward $R_{VAR}$ implemented? Is it a reward in VGPO algorithm or is it a training stage between SFT and VGPO?**
> We acknowledge that the original descriptions of how the visual-anchored reward $R_{VAR}$ is implemented (Appendix D.3 and Section 3.2 of the original submission) were not sufficiently clear.
> We have enhances these parts to present the implementation more clearly and in greater detail.
>
> VAR is implemented as a **reward model** trained using **softmax-DPO** on contrastive trajectory pairs generated in the MathCog-VAR subset. For each positive trajectory, we use an LLM to generate five negative trajectories by injecting one of the five structured reasoning-error types into the *Thinking* part, and all synthetic negatives are manually verified. Each trajectory is encoded into a hidden representation $h$, and VAR learns a scalar score $R_{\text{VAR}}(h) \in [0,1]$ that reflects its consistency with the visual evidence. During training, VAR is optimized with the softmax-DPO objective so that positive trajectories receive higher scores than their corresponding negative trajectories.
>
> After training, **VAR serves as an reward signal in the RL stage (VGPO) to detect reasoning drift and guide policy optimization**.
> We have expanded our description of the five common types of reasoning error incorporation in **Appendix D.3** and **Section 3.2** in our revised paper, which clearly illustrate how the VAR is implemented and how it is used in the CogFlow framework.
>
>
> ### **1.4 How is the correctness and format reward $R_{IR}$ implemented?**
> We apologize for the earlier ambiguity in our description of the correctness and format reward $R_{IR}$. Here's a detailed explanation of how the **correctness** and **format** rewards are implemented:
> 1. **Correctness Reward:**
>    The correctness reward, which is also known as **accuracy reward**, evaluates whether the model's response is correct. We assess correctness based solely on the `answer` part of the response. If the model's answer is correct, the reward is set to **1**, and **0** otherwise[1].
>
> 2. **Format Reward:**
>    The **format reward** verifies whether the response adheres to the required output schema. Specifically, the model must produce a **JSON-style** response, which includes tags such as:
>    `<Watching> ... </Watching> <Thinking> ... </Thinking> <Answer> ... </Answer>`.
>    If the response follows this structure, the format reward returns **1**, and **0** if the response does not comply.
>
> 3. **Inference Reward ($R_{IR}$):**
>    Finally, we combine the **accuracy reward** and **format reward** to compute the overall **inference reward** ($R_{IR}$), which is simply the sum of these two rewards:
>    $$
>    R_{IR} = R_{Acc} + R_{Fmt}
>    $$
>
> We have expanded our description of the correctness and format reward $R_{IR}$ implementation in **Appendix D.4** in the revised paper, which clearly illustrate how the $R_{IR}$ is implemented and how it is used in the CogFlow framework.

---

> ### Author Response · Authors · 2025-11-27
> **Rebuttal by Authors [3/4]**
>
> ### **1.5 How to compute the visual parameterized reward in the parameter space if the predicted primitive and the ground-truth primitive is not within the same class?**
>
> Firstly, we clarify that the Visual Parameterized Reward (VPR) is explicitly defined **within** each primitive class and does not compute distances across incompatible types. Concretely, as illustrated in Figure 3 and formalized in pseudocode below, we first restrict the primitives into three basic types in our datasets, namely **points, lines, and circles**, and then **perform a class-wise bipartite matching**: for each class, we construct a cost matrix in its own parameter space (for example coordinates for points, fitted line parameters for lines, center and radius for circles) and apply Hungarian matching to obtain an optimal one-to-one assignment between predicted and ground truth primitives of the same class. Primitives of different types (for example a predicted circle and a ground-truth ellipse) are never matched to one another and are instead treated as false positives or false negatives, which incur fixed penalties in the total VPR cost.
>
> We have expanded our description of the computation of visual parameterized reward in the parameter space if the predicted primitive and the ground-truth primitive is not within the same class in **Appendix D.2** and **Algorithm 1** in the revised paper, which clearly illustrate how the VPR is computed and how it is used in the CogFlow framework.
>
>   ```text
>   Algorithm: Multi-class Matching for VPR
>
>   Input:
>       Ground truth primitives G = {G_1, ..., G_m} with class labels in {point, line, circle}
>       Predicted primitives P = {P_1, ..., P_n} with class labels in {point, line, circle}
>
>   Output:
>       Optimal matching H ⊆ G × P
>       Total VPR cost S_VPR
>
>   1:  Initialize H ← ∅
>   2:  Initialize S_VPR ← 0
>   3:  Let K ← {point, line, circle}   # primitive classes
>
>   4:  For each class k in K do
>   5:      G_k ← { G_i ∈ G | class(G_i) = k }      # GT primitives of class k
>   6:      P_k ← { P_j ∈ P | class(P_j) = k }      # Predicted primitives of class k
>
>   7:      If |G_k| > 0 and |P_k| > 0 then
>   8:          Construct cost matrix C_k of size |G_k| × |P_k|
>   9:          For i = 1 to |G_k| do
>   10:             For j = 1 to |P_k| do
>   11:                 C_k[i, j] ← ‖ φ_k(G_k[i]) − φ_k(P_k[j]) ‖₂   # L2 in parameter space
>   12:             End For
>   13:         End For
>
>   14:         H_k ← Hungarian(C_k)           # optimal assignment for class k
>   15:         S_k ← sum_{(i,j) ∈ H_k} C_k[i, j]
>   16:         H ← H ∪ H_k
>   17:         S_VPR ← S_VPR + S_k
>
>   18:     Else if |G_k| > 0 then
>   19:         S_VPR ← S_VPR + λ_FN · |G_k|   # penalty for missed GT (false negatives)
>
>   20:     Else if |P_k| > 0 then
>   21:         S_VPR ← S_VPR + λ_FP · |P_k|   # penalty for false positives
>
>   22:     End If
>   23:  End For
>
>   24:  Return H, S_VPR
>   ```
>
> [1] Shao Z, Wang P, Zhu Q, et al. Deepseekmath: Pushing the limits of mathematical reasoning in open language models[J]. arXiv preprint arXiv:2402.03300, 2024.

---

> ### Author Response · Authors · 2025-11-27
> **Rebuttal by Authors [4/4]**
>
> ## **Weakness2: Fairness and analysis of the visual gate in Equation (4)**
> We thank the reviewer for this insightful comment.
> We believe that **our comparison is fair enough and the introduced visual gate achieves the best trade-off (significant performance improvement with minor extra costs)**. As suggested, we provide a comprehensive analysis to demonstrate the effectiveness and efficiency of the vision gate we introduced.
>
>
> ### **2.1 The impact of visual gate in the training phase**
> We acknoledge that our visual gate is equal to best-of-$k$ sampling if the threshold is not met in early attempts.
> However, we would like to highlight that **the visual gate incurs minimal overhead because it applies only to perception trajectories (regenerated up to $k$ times), whereas the reasoning trajectories are generated once**.
>
> On average across three training runs, **67.1% of tokens can pass the gate on the first attempt, showing that repeated perception generation is infrequent**.
> As evidenced in the last row and the fourth row from the bottom of Table 14 in the Appendix, **incorporating VGPO with the visual gate brings only an additional 1.3 hours of training time, which is fully acceptable for MLLM training**.
>
> ### **2.2 The impact of visual gate in the inference phase**
> Moreover, we add additional experiments to further analyze the impact of visual gate in inference phrase.
> Specifically, we compare three different inference schemes:
> - **single-pass**: model generates full trajectories ("Watching", "Thinking" and "Answer"), identical to standard MLLM inference;
> - **best-of-$k$ full**: the model generates $k$ full trajectories and selects the one whose final answer matches the ground truth. We note that this is not a realistic inference strategy, as it relies on ground-truth answers.
> - **visual gate (best-of-$k$ perception)**: CogFlow samples $k$ alternative perception trajectories, selects the one with the highest visual score, and then performs a single reasoning pass conditioned on the selected perception.
>
> To provide a fair comparison, we directly evaluate CogFlow and the open-source R1-style method (VLM-R1) under all three inference schemes using the same $k=3$. We additionally report the *average time required to process the 1,000 examples* for each setting.
>
>
> To allow a fair and controlled comparison, we evaluate CogFlow and the open-source R1-style method (VLM-R1) under all three inference schemes with a fixed $k=3$.
> We report the *average time required to process the 1,000 examples* for each setting.
> The results show that CogFlow outperforms VLM-R1 under *both* single-pass and best-of-$3$ full settings, demonstrating that **the performance gains are not attributable to more sampling**. Furthermore, our visual gate (best-of-$3$ perception) achieve nearly the same accuracy as best-of-$3$ full responses but with substantially lower inference time (2.72h vs.\ 3.06h vs.\ 5.52h). This confirms that **our comparison is fair and that CogFlow’s visual-gated design yields a more favorable computation–performance trade-off than full best-of-$k$ sampling.**
>
>
> | Model| **FlowVerse-CoT (%)** | **FlowVerse-Acc (%)** | Avg Time (h) |
> |------|-------------------|-------------------|-----------------|
> | VLM-R1-7B (single-pass) | 56.17 | 47.63 | 2.77 |
> | VLM-R1-7B (best-of-3 full) | 59.45 | 49.59 | 5.54 |
> | CogFlow-7B (single-pass) | 64.51 | 55.22 | 2.72 |
> | CogFlow-7B (best-of-3 full) | 66.04 | 56.17 | 5.52 |
> | CogFlow-7B (visual gate) | 66.03 | 56.24 | 3.06 |
>
>
> ### **2.3 The impact of $k$ in visual gate**
> Finally, we also add an ablation study on the choice of the sampling number $k$ in the visual gate. **This choice of $k$ is based on our preliminary experiments, which show that $k = 3$ provides a good balance between performance improvement and computational cost.** It is worth noting that $k = 1$ corresponds to the single-pass setting in the table above, since the single-pass variant does not require generating additional perception trajectories.
>
>
>
> | $k$| **FlowVerse-CoT (%)** | **FlowVerse-Acc (%)** | Avg Time (h) |
> |------|-------------------|-------------------|-----------------|
> | 1 | 64.51 | 55.22 | 2.72 |
> | 3 | 66.03 | 56.24 | 3.06 |
> | 5 | 66.36 | 56.55 | 3.71 |
>
> A more detailed discussion and the full quantitative table are provided in **Appendix E.6** and **Table 15**.

---

### Author Response · Authors · 2025-11-27
**Summary for Area Chairs**

Dear Area Chairs,

We sincerely thank all reviewers for their detailed evaluations and constructive suggestions. All concerns have been addressed point by point in the revised manuscript, where **the corresponding updates are highlighted in blue**.

Unfortunately, due to the technical issue in the ICLR 2026's review system, all the reviewers were unable to provide follow-up comments.
- Given that **most reviewers (3/4) initially assigned positive scores (6) while explicitly acknowledging the novelty and contribution of our work**, we believe that the clarifications and additional experiments presented in our rebuttal would further reinforce their already favorable assessment of the paper.

- For **the reviewer with an initial score of 4**, who positively highlighted our novel cognitive-inspired three-stage framework, the new dataset comprising three meaningful subsets, and the substantial performance improvements over baselines, **we have further provided comprehensive clarifications and in-depth analyses addressing their remaining concerns**. We believe that our responses adequately resolve the questions posed.

We understand that this unexpected situation has created additional workload, and we would like to express our sincere gratitude to the AC for the extra effort and care invested in handling our submission under these challenging circumstances. **To help reduce the burden on the Area Chairs, we would like to provide a summary to further highlight our contributions**
- All prior works neglect whether extracted visual cues are faithfully used in reasoning (i.e. reasoning drift). To address this issue, **we present CogFlow, a novel cognitive-inspired three-stage framework** that faithfully simulates the hierarchical human reasoning flow: perception ⇒ internalization ⇒ reasoning.
- In line with the human reasoning hierarchy, CogFlow holistically enhances all three stages:
  - **Synergistic Visual Rewards (SVRs)** complementarily enhance accurate and complete diagram perception in parametric space via the visual parameterized reward (VPR) and in semantic space via the visual semantic reward (VSR), thereby ensuring both local geometric fidelity and global perceptual coherence, which together form the foundation for effective visual mathematical reasoning;
  - **Visual-Anchored Reward (VAR)**  improves the knowledge internalization ability for promoting faithful conversion of perceptual outputs into a canonical context used for subsequent inference, serving as a reward signal in the RL stage to encourage the model to generate more faithful reasoning trajectories;
  - **Visual-Gated Policy Optimization (VGPO)** integrates visual gate mechanism with group-level optimization to filter high-quality perception trajectories and enhance the stability of reasoning.
- To support model training, **we curate a new dataset, MathCog**, comprising 100,000 examples (MathCog-SFT) for SFT phase, 10,000 examples (MathCog-RL) for RL phase, and 10,000 examples (MathCog-VAR) for VAR model training.

- **Comprehensive experiments on 6 visual mathematical benchmarks** validate that CogFlow achieves substantial gains in both answer accuracy and reasoning quality.

If there are any remaining concerns or unresolved issues, please let us know at your earliest convenience. We still have three days available to conduct additional experiments and will address any further questions or suggestions promptly.

---

### Author Response · Authors · 2025-12-01
**Summary of Rebuttal**

Dear Area Chair and Reviewers,

We sincerely appreciate all reviewers for their thoughtful assessment and constructive feedback on our submission. We are particularly grateful for the positive comments shared across the reviews, including:

- **Worthwhile research problem**: *addressing an important problem* (Reviewer `xQ8R`), *substantive contributions* and *a valuable addition to the field* (Reviewer `Wrhz`)
- **Novel perspective**: *a novel cognitive-inspired three-step framework* (Reviewer `gjDa`) and *conceptually clear and cognitively inspired framework* (Reviewer `joaQ`)
- **Comprehensive design**: *quite comprehensive method* (Reviewer `xQ8R`) and *a well-organized methodological design* (Reviewer `joaQ`)
- **Meaningful dataset**: *a new dataset featured with three subsets* (Reviewer `gjDa`), *a new dataset holding potential value for future research* (Reviewer `joaQ`) and *a complementary dataset* (Reviewer `Wrhz`)
- **Strong performance**: *substantial improvement compared with baseline methods* (Reviewer `gjDa`), *performing very well* (Reviewer `xQ8R`), *consistent improvements* (Reviewer `joaQ`) and *significant performance gains* (Reviewer `Wrhz`)

The primary concerns raised by reviewers relate to **missing implementation and dataset details**, as well as several **writing issues**. Reviewers also pointed out that CogFlow’s **generalization to additional benchmarks** (e.g., WeMath, LogicVista and DynaMath) had not been demonstrated, and that **more fine-grained ablation studies** were needed. We have addressed all concerns point-by-point, including those regarding implementation details, comprehensive evaluation, and additional comparisons. We have incorporated the following key updates into our revised manuscript:
1. **Comprehensive refinement of the overall narrative.**  We have thoroughly revised the introduction and methodology (**Sections 1 & 3**) to streamline the narrative and clarify the three-stage CogFlow pipeline, accompanied by improved **Figures 2–4** that enhance conceptual clarity and minimize visual ambiguity (as suggested by Reviewer `xQ8R`);
2. **Clearer description of MathCog construction.** We have moved and improved the figure illustrating the MathCog construction process to **Figure 10** in the main text, and expanded the detailed explanation of the dataset construction in **Appendix C.2.2** (as suggested by Reviewer `Wrhz`);
3. **More detailed description of all rewards.** We have refined the description of all rewards (including $R_{SVRs}$, $R_{VAR}$ and $R_{IR}$) in **Appendix D** and a concise specification of VPR and its matching procedure in **Appendix D.2** (as suggested by Reviewer `gjDa`);
4. **Additional benchmarking results.** We have added additional benchmarking results on WeMath, LogicVista and Dynamath datasets in **Table 4** (as suggested by Reviewers `xQ8R` and `joaQ`);
5. **Additional analysis on scalability and broad applicability.** We have evaluated CogFlow across different base models and model sizes in **Figure 16** (as suggested by Reviewer `joaQ`);
6. **Additional analysis of performance improvement sources.** With MathCog fixed as the training data, we illustrate the superiority of CogFlow over various post-training methods in **Table 14** (as suggested by Reviewer `Wrhz`).
7. **Additional analysis on multi-time perception generation in VGPO.** We compare CogFlow against multiple baselines across different generation–sampling schemes, confirming the fairness of the evaluation and the promising trade-off achieved by ours  in **Appendix E.3** and **Table 13**. (as suggested by Reviewer `gjDa`).

We thank all reviewers again for their constructive suggestions. All modifications are highlighted in different colors in the revised manuscript, and we welcome further discussion if needed.

Sincerely,

Authors of Submission 7237

---

### Meta-Review · Area_Chair_5N42 · 2026-01-12

**Summary:**

- Concerns were about missing implementation details and the fairness of the evaluation. The reviewer questioned whether the visual-gating mechanism constituted an unfair "best-of-k" sampling advantage over baselines.

- Concerns centered on presentation and scope. The writing was considered dense and jargon-heavy. The method's scalability beyond geometric shapes was questioned, and the evaluation was seen as limited, lacking tests on additional benchmarks like WeMath and LogicVista.

- Concerns were about generality and analysis. The evaluation lacked comparison with recent benchmarks, testing across different model scales/architectures, and a deeper analysis of why the multi-stage reward structure works.

- Concerns involved dataset transparency, ambiguity in the source of improvements, a perceived misalignment with the cognitive science framing, and missing experiments.

**Reviewer Concerns:**

Addressed Concerns:

- Missing Implementation Details. The authors provided clarifications for all requested details.

- Fairness of Visual Gating. The authors added an analysis comparing single-pass, best-of-k full trajectory, and their visual-gate (best-of-k perception) inference schemes.

- Limited Evaluation. The authors added new experimental results on three additional benchmarks (WeMath, LogicVista, DynaMath).

- Scalability Tests. The authors added experiments across different model sizes and two architecture families.

- Mechanism Analysis. The authors expanded the analysis in the revised manuscript to explain how each reward targets specific failure modes and how they work together, referencing new figures and ablation studies.

- Dataset. The rebuttal provides more detail on the data collection, filtering, annotation process, and the LLM-assisted generation of negative samples.

- Source of Improvement Ambiguity. The authors added an ablation study fixing the training data to MathCog and comparing different RL algorithms.

- Experiments. The authors added comparisons with SophiaVL-R1.

- Presentation. The authors committed to streamlining the narrative, reducing jargon, correcting typos and figure labels, fixing equations, and expanding figure captions.


Unaddressed Concerns:

- Cognitive Science Framing Misalignment. The rebuttal's explanation may not fully convince a reviewer who finds the analogy stretched, but it clarifies the authors' intended interpretation.

- Scalability to Non-Geometric/Natural Scenes. The concern about the method's core reliance on extractable geometric primitives for "visual mathematical reasoning" remains valid for its stated domain.

- Inference-Time Cost. The visual gate does add non-trivial inference time.

**Reviewer Scores:**

All reviewers are likely to retain their original scores.

---

### Decision · Program_Chairs · 2026-01-26

Accept (Poster)